# Optimizing the Cooling System of High-Speed Train Environmental Wind Tunnels Using the Gene-Directed Change Genetic Algorithm

**DOI:** 10.3390/e25101386

**Published:** 2023-09-27

**Authors:** Junjun Zhuang, Meng Liu, Hao Wu, Jun Wang

**Affiliations:** School of Aeronautic Science and Engineering, Beihang University, Beijing 100191, China; liumeng@buaa.edu.cn (M.L.); haowu@buaa.edu.cn (H.W.); wangjun@buaa.edu.cn (J.W.)

**Keywords:** optimizing the cooling system, exergoeconomic evaluation coefficient, genetic algorithm

## Abstract

Environmental wind tunnels play a crucial role in the research and development of high-speed railways. However, constructing and operating these wind tunnels requires significant resources, especially with respect to the cooling system, which serves as a vital subsystem. The cooling system utilizes an air compression refrigeration cycle and consists of multiple components. The efficient operation of these components, along with the adoption of appropriate strategies, greatly enhances the efficiency of the wind tunnel refrigeration system. Despite this, the existing methods for evaluating the refrigeration system do not fully capture the energy consumption of an air compression refrigeration system during practical use. To address this issue and effectively evaluate the wind tunnel refrigeration system, we propose using an exergoeconomic evaluation coefficient with experimental cycles to establish the system. This method incorporates the use of frequency coefficients and related parameters. By employing the newly developed evaluation coefficient as an objective function, we utilize the adaptive value-sharing congestion genetic algorithm to optimize the wind tunnel for high-speed trains. Furthermore, we compare the advantages and disadvantages of different optimization schemes. Traditional optimization methods prove inefficient because of the system’s numerous variables and the presence of multiple peaks in the objective function. Inspired by the biogenetic breeding method, we introduce an optimization approach based on a specific gene mutation. This innovative method significantly reduces optimization time and improves efficiency by approximately 17%.

## 1. Introduction

High-speed railways play an increasingly important role in economic transport. As an essential experimental facility for the research and development of high-speed railways, the high-speed railway environmental wind tunnel has received significant attention. Nowadays, some of the world’s famous environmental wind tunnels for trains include the Vienna Climatic Wind Tunnel, the CIRA Icing Wind Tunnel, the McKinley Climatic Laboratory, and the Jules Verne climatic wind tunnel. They have played a significant role in advancing high-speed train technology. One notable example is the Climatic Wind Tunnel in Vienna, which boasts an impeccable weather simulation system. China, the world’s foremost high-speed rail nation, is yet to possess its own high-speed rail wind tunnels. Since the foundation of The National Innovation Centre of High-Speed Train in 2019, China has sought to construct its own environmental wind tunnels. However, the construction and operation of high-speed railway wind tunnels for environmental testing, especially for full-scale models, incurs considerable costs. This is attributed to the vast dimensions and intricacy of the wind tunnels. The wind tunnel for high-speed trains is a large-scale experimental facility consisting of several systems, including power, refrigeration, control, measurement, data acquisition systems, and others. The refrigeration system, a crucial subsystem for wind tunnel operations, not only provides the cooling capacity to simulate a low-temperature environment in the experimental section of the wind tunnel but is also responsible for dissipating heat from other components in the tunnel. Because of the substantial demand for cooling, the system consumes a significant amount of energy. The refrigeration system comprises several components, including suction compressors, turbo-compressors, water coolers, return coolers, and turbo-expanders, resulting in a complex structure. The coordination of these components and the appropriate strategy for their use determine the advantages and disadvantages of the refrigeration system. Therefore, optimizing the design of the refrigeration system can effectively enhance its operating efficiency and reduce costs.

Before optimizing the wind tunnel refrigeration system, it is necessary to select or establish a suitable method for its evaluation. The methods of evaluating refrigeration systems have undergone many developments. Initially, most of them used the system energy analysis method, focusing on the cooling efficiency coefficient of performance (COP) and heat dissipation. These evaluations primarily studied the system’s energy use efficiency and consumption [1]. Wang et al. derived a general equation for the relationship between the refrigeration rate and cooling in a reheated air refrigeration cycle. They analyzed factors such as the degree of reheat return, the heat exchanger’s effectiveness, the compressor’s efficiency, and the expander’s efficiency, which influenced the refrigeration rate and coefficient. Chen et al. [2] further investigated the optimal distribution of thermal conductivity in the actual air reheat–cooling cycle and obtained the optimal cooling rate and coefficient. In addition to traditional objectives such as the refrigeration coefficient, Zhou et al. analyzed and optimized reversible [3] and irreversible [4] air reheat–refrigeration cycles under constant [5] and variable temperature [6] heat source conditions using the refrigeration rate density (the refrigeration rate divided by the maximum specific volume) as the thermodynamic optimization objective, and applied this methodology to optimize the cooling system for solar cells [7]. While the energy analysis method effectively describes changes in energy consumption, it fails to evaluate the system’s work capacity. Exergy analysis has been gradually incorporated into refrigeration system evaluations. Tu et al. [8,9] optimized the performance of an air reheat–cooling cycle under constant and variable temperature heat source conditions, using the cooling rate, ecological performance, and exergy efficiency as objective functions. They made a comprehensive comparison between the method of using a genetic algorithm to optimize the exergy efficiency and the traditional method of optimizing the cooling rate. An exergy analysis is a more comprehensive method of evaluation for refrigeration systems in terms of thermal performance alone. However, it does not consider the system’s economic characteristics or the construction cost required to improve its thermal characteristics. The exergoeconomic analysis method takes into account both the thermal performance and the construction and maintenance costs of a system while maximizing its thermodynamic performance. Tyagi et al. [10] studied and optimized a reheat irreversible refrigeration machine using energy economics, targeting the refrigeration rate per unit cost. Razmi [11] undertook an Exergoeconomic analysis of the refrigeration system and optimized the refrigeration system with the unit cost of refrigeration as one of the optimization objectives. Currently, the exergoeconomic analysis method is widely used in various thermal systems, such as solar-assisted power generation systems, hydropower cogeneration units, and gas–steam combined cycle units. In recent years, environmental considerations have been added to the evaluation of thermal systems. For example, Yasin [12] defined the thermal ecological objective ECOP (the cooling rate divided by the rate of usable energy loss) and used it to optimize air coolers. The evaluation of a thermal system now includes “energy, economy, and environment”. However, the refrigeration systems of high-speed train environmental wind tunnels possess some unique characteristics. This paper proposes a new evaluation index suitable for analyzing high-speed train environmental wind tunnels, as well as a method for evaluating their energy and economic performance based on experimental cycles. An adaptive value-sharing congestion genetic algorithm is employed to optimize the high-speed train environmental wind tunnel and compare different optimization schemes.

The environmental wind tunnel for high-speed trains must be tested under various experimental conditions, and a system model should be developed to analyze the operational mode. An in-depth analysis of the refrigeration system and wind tunnel operation under different operating conditions is crucial. Broniszewski [13] researched the impact of different fluid structures on wind tunnel resistance, and Cravero [14] further examined the correlation between fluid flow and heat. Because of the large number of components involved in the refrigeration system, uncertainty in the inputs should be considered when selecting parameters. Xia [15] examined the uncertainty related to the flow field input, while Cravero [16] carried out further analysis of the factors contributing to the uncertainty under varying flow conditions. Despite sensitivity analysis allowing for the removal of low-sensitivity design parameters in the refrigeration system, there are still numerous design parameters that necessitate optimization. Because of the large number of variables involved, traditional numerical solution methods are not feasible; thus, a genetic algorithm is chosen to optimally solve the cooling system. The genetic algorithm was first proposed by Professor Holland [17] in 1975 and has become a mainstream intelligent algorithm widely applied to thermodynamic problems [18,19,20,21]. However, this algorithm still has some shortcomings. For example, when the objective function has multiple peaks, the genetic algorithm tends to obtain local optimal solutions, leading to “genetic drift” [22]. Ajimakin et al. [23] proposed the “Genes Evolutionary Algorithm”, a binary algorithm that uses a binary function search space to avoid genetic drift by optimizing the fitness function. To improve the speed and accuracy of the solution, Giovanni et al. [24] refined the process of using genetic algorithms to solve the distributed and flexible job-shop scheduling problem by introducing a new local search-based operator in addition to traditional crossover and mutation operators. Jia Xu et al. [25] introduced the gene sequence comparison method from biology to improve the crossover operation, resulting in a faster solution speed and more stable results.

In the process of optimizing high-speed train environmental wind tunnels, the system involves numerous variables, and the objective function has many peaks. The existing optimization methods require substantial lengths of time for optimization. To address these issues, a specific gene-mutation-based optimization method is proposed in addition to the improved original genetic algorithm. This method significantly reduces optimization time and improves efficiency.

## 2. Exergoeconomic Analysis with an Experimental Cycle

The cooling system of a high-speed train environmental wind tunnel is based on the Brayton inverse cycle and comprises specific components, as shown in Figure 1. The system follows the Brayton cycle in which air is compressed and expanded to generate low-temperature cooling air for the high-speed train environmental wind tunnel. The main components include the inlet compressor, turbo-compressor, turbo-expander, water cooler, return cooler, and cooling tower.

The procedure begins with the compression of external air by the inlet compressor. The compressed air is then dried and purified, resulting in a high-temperature, high-pressure dry clean gas. After further compression via the turbo-compressor, the air pressure and temperature increase. Through the cooling system, the temperature of the compressed air is reduced by passing it through the water cooler and the re-cooler. The cooled, compressed air then enters the turbo-expander, where expansion produces low-temperature gases at atmospheric pressure. This cooled air is then sent into the high-speed train environmental wind tunnel via a pump serving as the cold source. The power output from the expander can be utilized to drive the co-axial compressor, enabling the recovery of the compressed air’s energy and increasing the energy efficiency of the system.

Compared with traditional refrigeration systems, the refrigeration of a high-speed train environmental wind tunnel possesses two distinct characteristics.

First, since the high-speed train environmental wind tunnel needs to simulate various environmental conditions, the cooling capacity and temperature of the cooling source provided by the refrigeration system vary accordingly. Additionally, the refrigeration system operates in different states. Existing thermodynamic evaluation systems often focus on assessing the performance of a single operating state, selecting one or several specific states for evaluation. However, in an air compression refrigeration system, each state is equally important, and evaluating only a few results in an incomplete evaluation of the train environment system. Moreover, the same set of equipment is used for the different states in the refrigeration system, so the design of the system components should consider the most severe environment to ensure that the cooling needs are met for all experimental sections of the high-speed train environmental wind tunnel.

Second, since the air compression refrigeration system extracts a certain amount of air from the environmental wind tunnel and returns it to the cooler as the main cold source, the airflow in the latter half of the wind tunnel becomes smaller. The airflow slows down as a result of the constant air density, leading to a reduction in flow loss in the wind tunnel. In the environmental wind tunnel, this pumping method has a minor impact on the experimental section with higher wind speeds but a more significant impact on the experimental section with lower wind speeds. The original methods of evaluating refrigeration systems paid little attention to the energy flow coupling between the refrigeration and other systems and failed to accurately represent the influence of changes in the refrigeration subsystems on the operational efficiency and cost of the complete external system.

To address these two problems, improvements based on the original economic analysis and evaluation can be made. For instance, environmental wind tunnels for high-speed trains primarily serve various environmental experiments. Our research found that testing a high-speed train, from the pre-research stage and initial testing to the final validation experiments, product finalization, and mass production, requires undertaking a series of experimental procedures. This entire experimental process follows a well-established set of validation standards. It can be assumed that there exists a complete experimental cycle for train experiments applicable to most environmental conditions and trains. Therefore, the weight coefficient of the system usage frequency $T_{i}$ can be increased using economic success to describe the frequency of environmental wind tunnel usage in different states during the complete cycle of a high-speed train environmental experiment. In this way, a new system of environmental wind tunnel evaluation based on the experimental cycle of a high-speed railway train can be formed. This new system of evaluation can be expressed using Equation (1).
(1)$$F_{cyc,}{}_{k}=\sum \left(C_{out,k}\cdot T_{k}{}_{{}_{i}}\right)=\sum \left(C_{in,k}\cdot T_{k}{}_{{}_{i}}\right)+Z_{k}$$

Alternatively, Equation (2) can be obtained from Equation (1):(2)$$F_{cyc,}{}_{sys}=\sum F_{cyc,}{}_{k}=\sum_{k}\left(\sum_{i}\left(C_{in,k}\cdot T_{k}{}_{{}_{i}}\right)+Z_{k}\right)$$
where $F_{cyc,}{}_{sys}$ represents the economic cost of the experimental cycle, USD/h; $F_{cyc,}{}_{k}$ represents the economic cost of the component in the experimental cycle, USD/h; $T_{k}{}_{{}_{i}}$ denotes the weighting coefficients for different experimental states in the full-speed cycle of the train experiment. These coefficients are calculated to obtain the experimental cycle weights and can be expressed using Equation (3).
(3)$$T_{k}{}_{{}_{i}}=T_{T}\cdot T_{V}={\left[T_{h}T_{c}\right]}^{T}\times \left[T_{hs}T_{ms}T_{ls}\right]$$

The impact of the refrigeration system’s energy consumption on the other systems in the wind tunnel can be addressed by conducting an exergy analysis on the entire experimental system. While this method accurately analyzes the influence of changes in the refrigeration system on other systems, it also significantly increases the number of analysis systems, thereby increasing the analysis cost. Moreover, it may fail to highlight the energy flow characteristics of the refrigeration system as a result of excessive redundant data, thus complicating the discovery of the refrigeration system’s energy flow coupling law. To address these related problems, a detailed energy flow analysis of the complete system of the high-speed train environmental wind tunnel is conducted first to clarify the energy flow coupling between the refrigeration system and the other systems in a high-speed-train environmental wind tunnel. Subsequently, we analyze and quantify the exergoeconomic cost of the refrigeration system in relation to the other systems present in the wind tunnel *V_Os_*, USD/h. *V_Os_* and the previous *F_sys_* comprise the final evaluation factor for the refrigeration system known as *G_sys_*, USD/h. Equation (4) demonstrates how *G_sys_* can be rewritten to determine its final form
(4)$$G_{sys}=F_{cyc,}{}_{sys}+V_{Os}=\sum F_{cyc,}{}_{k}+\sum \left(E_{Os}\cdot T_{k}{}_{{}_{i}}\right)=\sum_{k}\left(\sum_{i}\left(C_{in,k}\cdot T_{k}{}_{{}_{i}}\right)+Z_{k}\right)+\sum \left(E_{Os}\cdot T_{k}{}_{{}_{i}}\right)$$
where $E_{Os}$ represents the external impact values of the systems in USD/h.

This evaluation index enables a more effective comparison of the advantages and disadvantages of different environmental cooling systems for high-speed trains. Consequently, it serves as the main objective function for the subsequent optimization of environmental cooling.

## 3. Optimization Model for Air Compression Refrigeration Systems

By considering the principles of refrigeration, exergoeconomic analyses, and other methods, we developed an evaluation coefficient $Z_{sys}$ defined by Equation (5) for the air compression refrigeration system used in high-speed-train environmental wind tunnels
(5)$$\min\left(G_{sys}\right)=\sum_{k}\left(\sum_{i}\left(C_{in,k}\cdot T_{k}{}_{{}_{i}}\right)+Z_{k}\right)+\sum \left(E_{Os}\cdot T_{k}{}_{{}_{i}}\right)$$

The modeling and sensitivity analyses of design parameters for air compression refrigeration systems have been discussed in [26]. Optimization function variables can be categorized into two main groups: system component selection indexes and operating parameter indexes. The key system component selection indexes for optimization are summarized below in Table 1.

The operating parameter indexes are summarized in Table 2 below.

The refrigeration system consists of six operating states, each corresponding to three operating parameter indicators. Therefore, a total of 18 operating parameter indicators must be optimized. Additionally, three other constraints need to be considered for optimizing the system:$$\left\{\begin{array}{l} \varepsilon_{\min}\leq \varepsilon_{i}\leq \varepsilon_{\max}\\ {\dot{m}}_{\min}\leq {\dot{m}}_{i}\leq {\dot{m}}_{\max}\\ q_{shi_{i}}\geq q_{xu} \end{array}\right.$$

The compressor’s compression ratio in the system must exceed the minimum compression ratio and be less than the specified maximum. Simultaneously, the mass flow rate of the system must fall within the authorized range of each component’s mass flow rate. Ultimately, the system should deliver refrigeration capacity that meets or surpasses its requirements.

We modeled the operation of the refrigeration system in order to calculate its performance in various states. In the refrigeration system model, the turbine and compressor are analyzed using one-dimensional flow characteristic analysis and the principle of similarity. The specific modeling process is shown in Figure 2.

The re-cooler is modeled using the effective number of heat transfer units and the mean temperature-difference design method classical approach. The specific modeling process is shown in Figure 3.

The key system components and operating parameters are then optimized in turn to obtain the exergoeconomic cost. The objective function is obtained by multiplying the result with the frequency coefficients of the wind tunnel and adding the effect on the other parameters.

This article employs two primary optimization algorithms: the fitness-sharing crowding genetic algorithm and the gene-directed change genetic algorithm. The latter is an improvement of our existing algorithm based on the thermodynamic laws of the system. A detailed description of the gene-directed change genetic algorithm will be provided in later sections. The fitness-sharing crowding genetic algorithm has gained popularity as a reliable method for system optimization in recent years. It serves as the primary approach to addressing the issue of “genetic drift” in multipeak functions. The fundamental concept behind the adaptive value-sharing crowding genetic algorithm involves combining the ideas of adaptive value sharing and crowding. Specifically, the adaptive value-sharing genetic algorithm incorporates the sharing concept before the selection process, while the replacement process completely replaces all old individuals with new ones. On the other hand, the crowding algorithm employs non-dominated sorting to classify the population and calculate the crowding distances of individuals to maintain population diversity. The adaptive value-sharing and crowding genetic algorithm adjusts the adaptive values of individuals during the genetic algorithm’s selection process and introduces competitive comparison in the replacement process. Combined, these two algorithms complement each other, enhance the algorithm’s search capability, and resolve issues associated with “genetic drift” and similar problems. The general optimization steps are illustrated below in Figure 4.

Firstly, an initial parent population Gen = 1, size N is randomly generated. Then, the parent population generates an offspring population of the same size. The parent and offspring populations are merged to form a population size 2N. This newly generated population undergoes non-dominated fast sorting. The crowding degree of all individuals in the 2N-sized population is computed, and appropriate individuals are selected using the crowding degree relationship and individual crowding degrees to form a new size N population. Traditional genetic algorithms, including crossover operations, are then used to generate a population that repeats the above steps. By considering the crowding relationship and the crowding sizes of individuals, suitable individuals are selected to form a new parent population of size N. Through crossover and mutation, an offspring population of size N is generated and merged with its parent population. Non-dominated fast sorting is repeated until the number of generations reaches the predetermined value. The algorithm possesses the following advantages:The implementation of a fast non-dominated sorting algorithm significantly reduces the complexity of the optimization algorithm;The introduction of an elite conservation mechanism ensures that good individuals are not lost during the evolutionary process by allowing the parent population to compete with their offspring to produce the next generation;Using crowding and crowding comparison operators promotes population homogeneity and diversity preservation.

### 3.1. Refrigeration Optimization Program

Three model optimization scenarios were developed as follows:

Option 1. The optimization of refrigeration systems based on the most severe usage conditions.

The mainstream approach for optimizing refrigeration systems is to use a scheme based on the most stringent boundary conditions. This method ensures that the component parameters of the selected system can meet the needs of all other cooling states, thus ensuring system stability under various operating conditions. Additionally, optimizing the system under harsh environmental conditions that often coincide with high energy consumption can effectively reduce system costs. The key operational ideas of this system are as follows in Figure 5:

Initially, the general system is optimized for operations at a temperature of −70 °C and a wind speed of 100 m/s in the experimental section of the environmental wind tunnel. The optimization parameters include equipment selection and wind tunnel operation parameters under this condition. To obtain credible results and compare the efficiencies of different methods, the termination conditions of the genetic algorithm are employed to determine if the desired results are achieved. The algorithm terminates when the system population remains unchanged or only changes slightly. Generally, a region is defined, and the optimization process continues until all population members are within that region for a certain number of generations. If the difference between 20 consecutive genetic generations is less than 1‰ of the minimum value, the system is considered stable and reaches termination condition. The system stability condition is determined using specific coefficients.
$$\left(F_{\left(Gen-20\right)}{\left(x\right)}_{\max}-F_{Gen}{\left(x\right)}_{\min}\right)\leq F_{Gen}{\left(x\right)}_{\min}\times 1‰$$

The optimal scheme for the stabilized system is determined as the minimum value, and the system design parameters are sequentially applied to different systems to carry out the system optimization. This process yields a control scheme for the air compression refrigeration system under different conditions. The relevant scheme parameters are sorted out, and evaluation coefficients for high-speed trains’ environmental wind tunnel cooling in different states are calculated.

Option 2. Optimization using the cooling evaluation factor as an objective function.

In contrast to the first option, which derives wind tunnel evaluation coefficients indirectly, the second option directly utilizes refrigeration evaluation coefficients as functions of the objective coefficients. The approach is presented in Figure 6:

Compared with Option 1, Option 2 involves fewer steps but significantly increases the number of variables to be optimized from 9 to 24. Consequently, there is a substantial increase in the number of system optimization algorithms, resulting in a longer computation time. This phenomenon arises because, during system optimization, multiple systems need to be computed simultaneously. If any of these systems fail to meet the requirements, the optimization solution is deemed noncompliant, leading to a large number of potential optimization solutions that cannot be executed and consequently slowing down the convergence of system results. To address this issue, improvement schemes are proposed for different systems to enhance the feasibility of obtaining effective solutions and improve the convergence speed of the optimization system. This approach is particularly effective in precomputed systems.

Option 3. Refrigeration systems for two sets of equipment.

The previous study focused solely on analyzing a single refrigeration system. However, in practical applications, safety hazards and power underutilization can occur because of issues such as the relative size of the compressor. To address these concerns, it is possible to employ two refrigeration systems operating in parallel, which can effectively resolve these issues. The approach is shown in Figure 7.

The system optimization approach for Option 3 remains the same as in Option 2. The only difference is the inclusion of an extra set of refrigeration equipment. The solving process follows the same methodology as Option 2.

### 3.2. Gene-Directed Change Genetic Algorithm

Because of the addition of a refrigeration system in Option 3, the number of variables to be optimized doubles from 24 to 48. This increase in system variables significantly raises the complexity of system optimization. To address these challenges, we propose a genetic optimization algorithm using fixed-point gene mutation.

Gene point mutation techniques involve introducing favorable changes into the target DNA fragment or genome to achieve desired characteristics. Polymerase chain reactions (PCRs) are commonly used to facilitate base addition, deletion, point mutation, and other modifications. These techniques have proven to be valuable tools in genetic research across various fields. In medicine, they have been used to correct genetic diseases and treat cancer and other illnesses, and in agriculture, to develop disease-resistant and insect-resistant seeds. Using fixed-point mutation technology has significantly reduced breeding time and improved efficiency.

The current genetic algorithms are derived from biologically relevant features and primarily rely on crossover and random mutation to generate offspring. In the refrigeration system optimization process, we employed genetic information crossover and random mutation. However, because of the strong coupling between system variables, crossing two excellent offspring often fails to produce equally excellent offspring. Consequently, obtaining superior offspring through crossover alone becomes difficult. On the other hand, random mutation offers a relatively high chance of generating positive effects in the later stages of refrigeration system optimization. However, during the initial phases, the disorderly nature of evolution leads to an overall low system efficiency. By using fixed-point gene mutation technology, we can effectively address this issue. The specific operational steps are presented in Figure 8:

The gene-targeting mutation strategy is as follows:Reduce the dimensional design parameters of all components (excluding the compressor and turbine inversions) and system control variables by 5%;Maintain the refrigeration 1 system component parameters and control variables as they are but reduce the refrigeration 2 system component parameters and control variables by 5%;Keep the parameters of all components unchanged while reducing all state system control variables by 5%;Keep the parameters of all components unchanged. Reduce the system control variables in states 1 and 4 (high-speed states) by 5% while leaving the system control quantities in the remaining states unchanged;Keep the parameters of all components unchanged. Reduce the system control variables in states 2 and 5 (medium-speed states) by 5% while leaving the system control quantities in the remaining states unchanged;Keep the parameters of all components unchanged. Reduce the system control variables in states 3 and 6 (low-speed states) by 5% while leaving the system control quantities in the remaining states unchanged;Keep the parameters of all components unchanged. Reduce the system control variables in states 1, 2, and 3 (low-temperature states) by 5% while leaving the system control quantities in the remaining states unchanged;Keep the parameters of all components unchanged. Reduce the system control variables in states 4, 5, and 6 (room-temperature states) by 5% while leaving the system control quantities in the remaining states unchanged;Keep the parameters of all components unchanged. Maintain the control variables of System I while scaling down the control variables of all states of System II by 5%;Keep the parameters of all components unchanged. Maintain the control variables of System I while reducing the system control variables of states 1 and 4 (high-speed states) of System II by 5%. Leave the system control quantities for the remaining states unchanged;Keep the parameters of all components unchanged. Maintain the control variables of System I while reducing the system control variables of states II and V (medium-speed states) of System II by 5%. Leave the system control quantities for the remaining states unchangedKeep the parameters of all components unchanged. Maintain the control variables of System I while reducing the system control variables of states III and VI (low-speed states) of System II by 5%. Leave the system control quantities for the remaining states unchanged;Keep the parameters of all components unchanged. Maintain the control variables of System I while reducing the system control variables of states I, II, and III (low-temperature states) of System II by 5%. Leave the system control quantities for the remaining states unchanged;Keep the parameters of all components unchanged. Maintain the control variables of System I while reducing the system control variables of states IV, V, and VI (ambient states) of System II by 5%. Leave the system control quantities for the remaining states unchanged.

Each row of the matrix represents a feasible direction for the fixed-point mutation. We can draw an analogy with biological fixed-point mutations to conclude that fixed-point mutations have a higher probability of producing improved individuals, thereby accelerating population evolution. Furthermore, since the best individual in the population has a greater chance of being optimized than other individuals, the number of systematically superior individuals in the population rapidly increases, leading to faster convergence. However, using fixed-point mutations also presents challenges, such as reducing population diversity, which results in certain information from the parent generation not being passed on to the offspring, leading to “genetic drift.” To mitigate this, it is necessary to expand the population size and increase the number of parents.

## 4. Results and Discussion

We utilize the three different optimization schemes mentioned earlier to optimize the design of the cooling system used in high-speed train environmental wind tunnels. The goal is to determine cooling schemes for the wind tunnel under different operating conditions. Additionally, we compare the characteristics of the system optimization process using traditional optimization methods with those that use the genetic method employing a fixed-point mutation.

### 4.1. Comparison of Refrigeration System Optimization Results

According to industry standards and engineering expertise, the experimental requirements for high-speed train testing generally follow a 1:3:1 time ratio of high speed/conventional speed/low speed. The distribution of experimental conditions between high-temperature, medium-temperature, and low-temperature states should be approximately the same. However, since the high-temperature portion of the wind tunnel can be cooled by the tunnel itself, we only need to consider the experimental requirements for the low- and room-temperature sections. The specific experimental requirements for each section and the frequency coefficients for the wind tunnel are shown in Table 3:

The primary cooling demand in the system is influenced by the wind pressure, which is determined by the wind speed requirement in the experimental section. Additionally, there is a fixed cooling demand of 2 MW in the gas section, and the heat dissipation in the system depends on the temperature in the experimental section.

First, the three methods are compared on the basis of the cooling evaluation coefficient per unit of time, as shown in Figure 9. This index provides an intuitive reflection of the system’s cooling cost. From the comparison, we can observe that the second method performs better than the first, with a reduction in the coefficient of approximately USD 183.12 per hour, resulting in a 5% decrease in the system’s cost per unit of time. This indicates that by adopting different system selection methods, the system cost can be effectively reduced. The cost of the third method is significantly lower than the first two, with a reduction in the coefficient of approximately USD 1724.78 per hour, representing cost savings of about 43%. Therefore, it can be concluded that adding a completely independent system is a very effective approach to reducing the system’s cost index. However, further analysis is required to understand the reasons behind these cost reductions.

Figure 10 provides a comparison of the operating efficiencies and amounts of refrigeration power for the different options in various states. From the graph, it is evident that the primary factors contributing to the difference in system operating costs are the high levels of energy consumption and low levels of refrigeration efficiency in the low- and medium-speed states. Although method 1 shows a slightly better efficiency level than the other two methods at low temperatures and high speeds, it fails to compensate for the high refrigeration consumption in other states. On the other hand, method 3 demonstrates significantly lower refrigeration costs than the other two methods, primarily because of its improved efficiency at medium and low speeds. This can be attributed to the fact that when there is only one system, the refrigeration equipment flow needs to be larger to ensure cooling requirements are met during extreme conditions. However, the cooling capacity of the equipment is limited, resulting in excess cooling at lower temperatures. In order to maintain the required temperature in the experimental section of the wind tunnel without excessively lowering it, the surplus low-temperature air must be discharged after re-cooling, leading to a loss in system efficiency.

Table 4 provides a comparison of the construction costs for the different system options. The results show that the total construction costs of the three systems are relatively similar, with Option 2 being the lowest. This is primarily due to the smaller size of the required system, which results in a reduced cooling margin during system operation. The higher construction cost of System III can be attributed to the inclusion of two independent refrigeration units. Although each individual unit has a relatively low cost, the combined cost of both units is approximately 5% higher than the construction cost of the other two systems. In Option 3, the two refrigeration systems are not of equal size. By analyzing previous usage patterns, it has been determined that one of the refrigeration systems in Option 3 is suitable for the wind tunnel under medium- and low-speed conditions, while the other system is appropriate for medium- and low-speed conditions. It is important to note that in practical engineering scenarios, the construction cost of Option 3 may be higher. This is because additional valves and interconnections may be required between the two systems, in addition to an increase in floor space and other ancillary equipment. Despite this, the increase in the cost of construction is significantly smaller than the energy cost savings over the system’s lifespan.

The influence of the refrigeration system on the energy consumption of the other systems is illustrated in Figure 11. As mentioned earlier, although the refrigeration system does influence the operational status of the other systems, it is generally not considered a major factor in the actual design and construction process. Hence, it can be assumed that the construction costs of the other systems are not affected by the refrigeration system. Regarding power, positive values indicate performance enhancements for other systems, while negative values imply reductions. It is evident that the number of external influences is relatively small compared with the overall energy consumption of the refrigeration system. The power impact is higher for the high-speed system and relatively low for the medium- and low-speed systems. Within the medium- and low-speed states, the impact is greater at lower temperatures. This is due to the fact that a significant portion of the gas extracted from the system is mixed with low-temperature gas and re-entered into the wind tunnel. The system’s enhanced refrigeration capacity at low temperatures enables more air to be pumped out, resulting in a larger reduction in energy costs for the remaining part of the system.

### 4.2. Comparison of Cooling Optimization Methods

In this paper, the process was programmed using MATLAB 2018b, based on the improved algorithm mentioned earlier, to optimize and analyze the method of selection and mode of operation of the high-speed train environmental wind tunnel. To verify the effectiveness and feasibility of the second method and compare the running speeds of the two algorithms, each algorithm was independently executed 10 times on a computer system, using an Intel(R) Core(TM) i7-3770K CPU @ 3.50 GHz and 16 GB RAM. Simulink’s model was used to run the wind tunnel refrigeration models. The algorithm parameters were as follows: the population size for optimization was set to 48, the number of populations was set to 200, the stability of the population was determined using a minimum of two elements, and the change in value between optimal solutions for five consecutive generations was ≤0.01. Additionally, the coefficient of variation within the same generation, calculated as {(maximum value − minimum value)/minimum value}, was ≤1‰.

Since the initial population impacts the number of iterations, the same population was used for both methods. For the mutation matrix, the gene-targeting mutation strategy was employed, as shown below in Figure 12:

Using the two methods, 10 optimization calculations were performed for the air compression refrigeration cycle. The optimal solutions and the number of optimization generations for each dataset are presented in Table 5. It can be observed that the final optimized objectives using the two methods are very close to each other, indicating the effectiveness of method 2 for the calculations. In terms of computational efficiency, it is noted that program 1 required an average of 984.4 iterations with a computation time of 37.69 h, whereas program 2 required an average of 812.9 iterations with a computation time of 31.12 h. This represents an approximate 17% increase in computational speed. Although the accuracy of the calculations between the two methods is similar, using method 2 significantly enhances computational efficiency and reduces computation time.

Taking the sixth calculation of both methods as an example, this section analyzes the advantages and disadvantages of the genetic optimization algorithm, which uses a gene fixation mutation, in detail. A convergence iteration diagram of the objective function is shown in Figure 13 and Figure 14. From the graph, it can be observed that both methods exhibit slower convergence in the later stages. However, during the initial stage of the iterative experiment, it is evident that the new method demonstrates a significantly better convergence speed compared with the original method. The improved convergence speed in the early stage can be attributed to the utilization of system transformation in which the system’s change pattern becomes more apparent. By employing the gene transformation method based on fixed-point mutation technology, specific independent variables can be precisely altered, thus avoiding the unintended consequences of gene mutation and reducing meaningless system changes. Consequently, this approach enhances the generation of superior individuals and accelerates the convergence iteration speed of the system.

However, as the iterative population reaches the later stages in which the system’s change pattern is less obvious and there are closer coupling relationships between different independent variables, the use of fixed-point gene changes may fail to meet the operational requirements of the system. These individuals are directly eliminated. At this point, the generation of excellent individuals in the population primarily relies on gene mutation to achieve optimization efficiency. As a result, the computational efficiencies of both methods do not differ significantly during the late iterations.

In summary, using genetic algorithms with fixed-point mutations can improve the optimization efficiency in the initial stages of the system more rapidly without significantly impacting the optimization results of the system in the long run.

## 5. Conclusions

This paper focuses on optimizing the refrigeration system of a high-speed train environmental wind tunnel. Before optimizing the refrigeration system, it is necessary to determine the optimization objective. Existing refrigeration system evaluation indexes mostly apply to stable operation systems, lacking evaluation methods for systems with complex working conditions. Therefore, a new system of evaluation for a refrigeration system based on the system’s energy and economic performance is proposed. The selection and operation scheme of the refrigeration system is modeled using a genetic algorithm. The operating characteristics of the system under different optimization methods are compared. Finally, an improved genetic optimization method for refrigeration systems is proposed, which significantly improves the system’s optimization efficiency without compromising operating accuracy.

The key conclusions are as follows:Using the original exergoeconomic analysis method, a weight coefficient for the frequency of different states in the refrigeration system is introduced to evaluate the comprehensive energy consumption of the high-speed refrigeration system under complex working conditions. Additionally, by analyzing the coupling relationship between the refrigeration system and other systems in the train’s wind tunnel, the impact cost of the refrigeration system in the wind tunnel system is introduced. This cost reflects the coupling relationship between the refrigeration system and the other systems during operation and is incorporated into the new method of evaluating refrigeration systems via energy prices. Ultimately, a system for evaluating a refrigeration system based on the system’s energy and economic performance is developed.A congestive genetic algorithm with shared adaptation values is employed to optimize the system under different schemes. It is observed that although Option 3 has inferior construction costs and cooling efficiencies at low temperatures and high speeds compared with the other two schemes, it outperforms the other two schemes overall because of its excellent performance under ambient operating conditions. The energy cost of Option 3 per unit of time is approximately 2204.81 USD/h, which is about 43% lower than that of Option 3, resulting in a cost saving of around 1724.78 USD/h.An improved genetic optimization algorithm using fixed-point gene mutation is proposed. By comparing the two algorithms, it is found that the new method enhances the optimization efficiency of the genetic algorithm in the early stages of the system. This leads to a reduction in the number of optimization iterations from 984.4 to 812.9 generations and a decrease in the average computation time from 37.69 to 31.12 h, resulting in a 17% increase in computational efficiency.

## Figures and Tables

**Figure 1 entropy-25-01386-f001:**
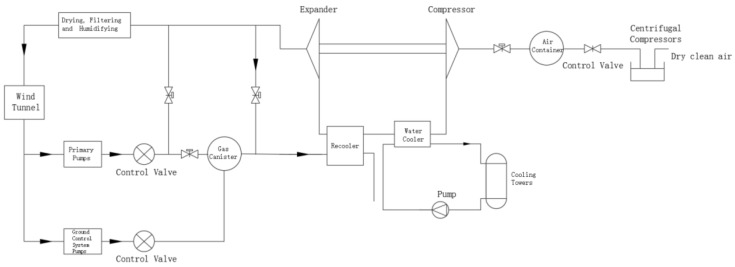
Air compression refrigeration cycle.

**Figure 2 entropy-25-01386-f002:**
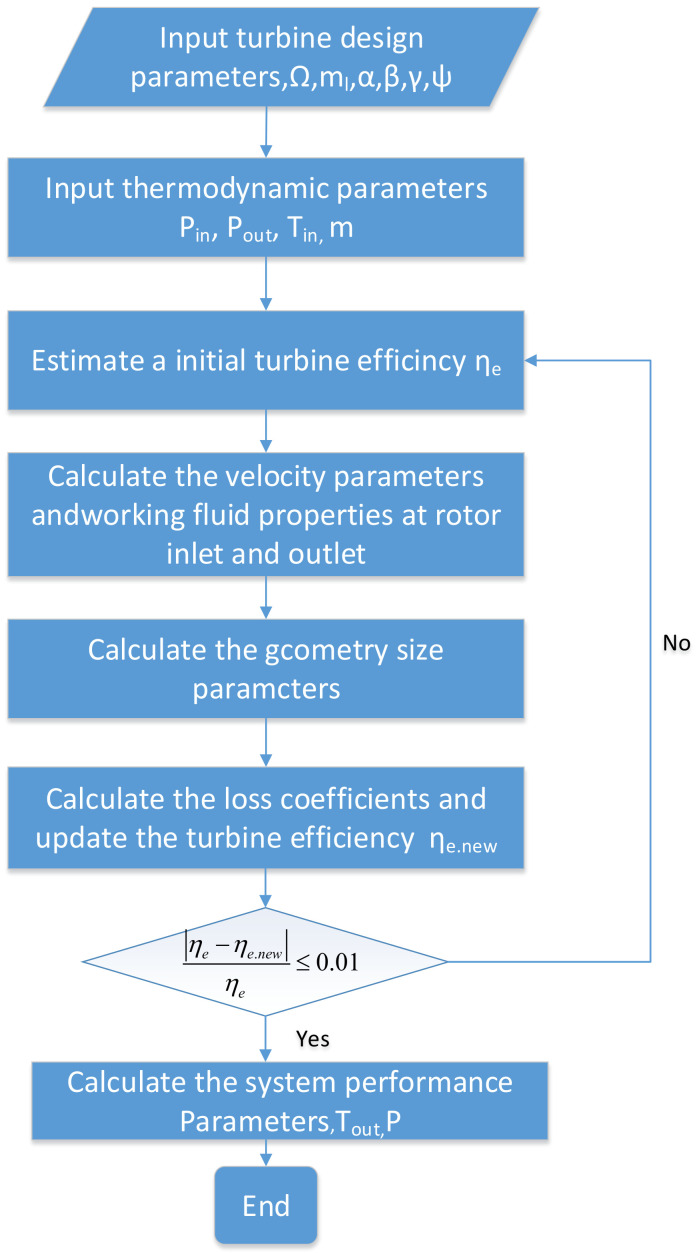
Turbine modeling flowchart.

**Figure 3 entropy-25-01386-f003:**
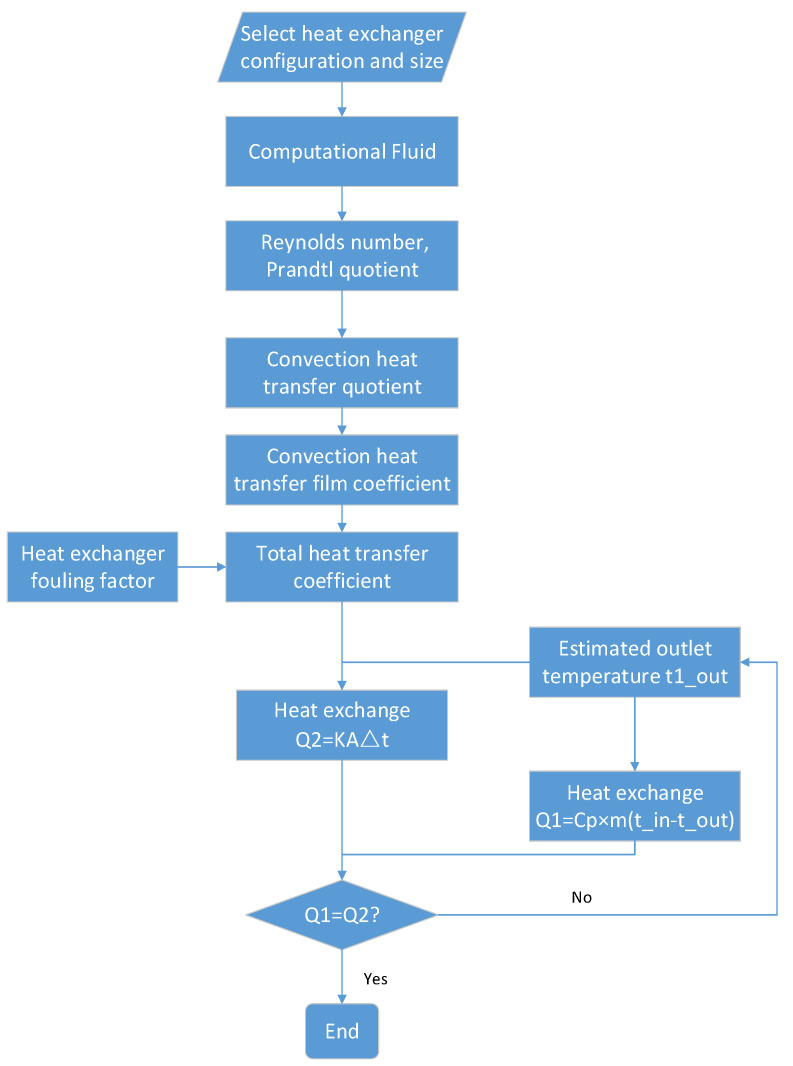
Heat exchanger modeling flowchart.

**Figure 4 entropy-25-01386-f004:**
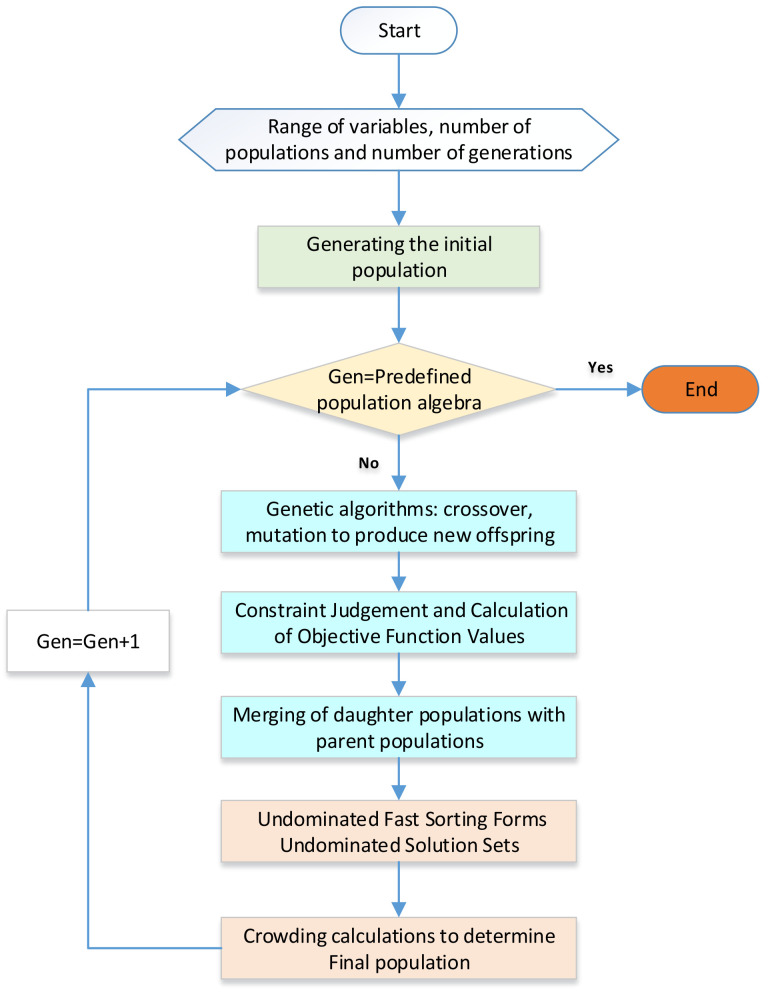
The fitness-sharing crowding genetic algorithm.

**Figure 5 entropy-25-01386-f005:**
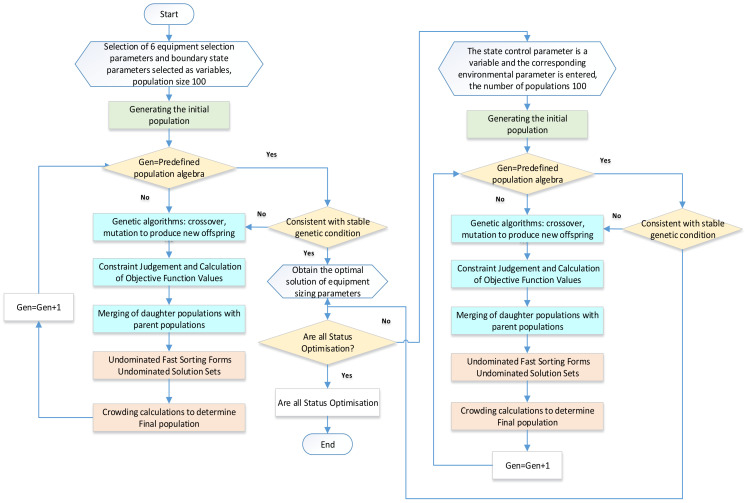
Option 1. The optimization of refrigeration systems based on the most severe usage conditions.

**Figure 6 entropy-25-01386-f006:**
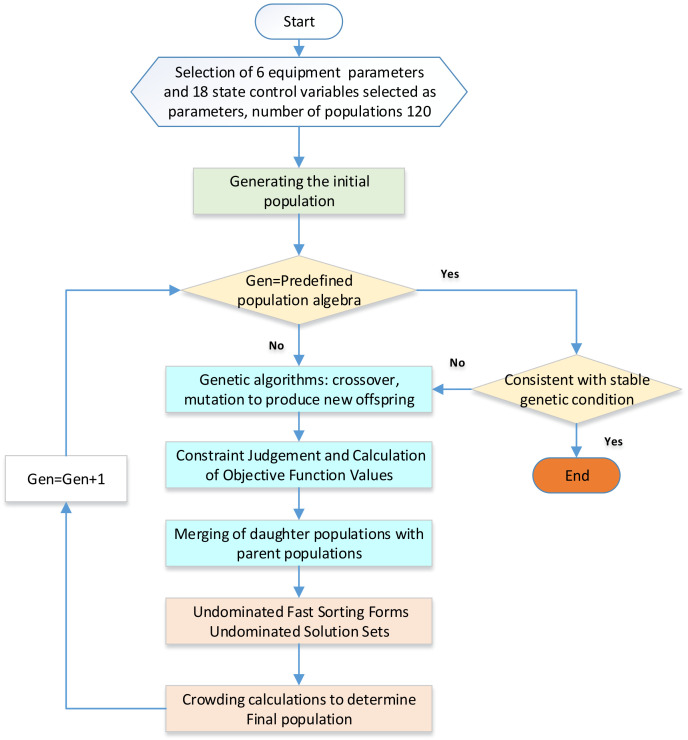
Option 2. Optimization using the cooling evaluation factor as an objective function.

**Figure 7 entropy-25-01386-f007:**
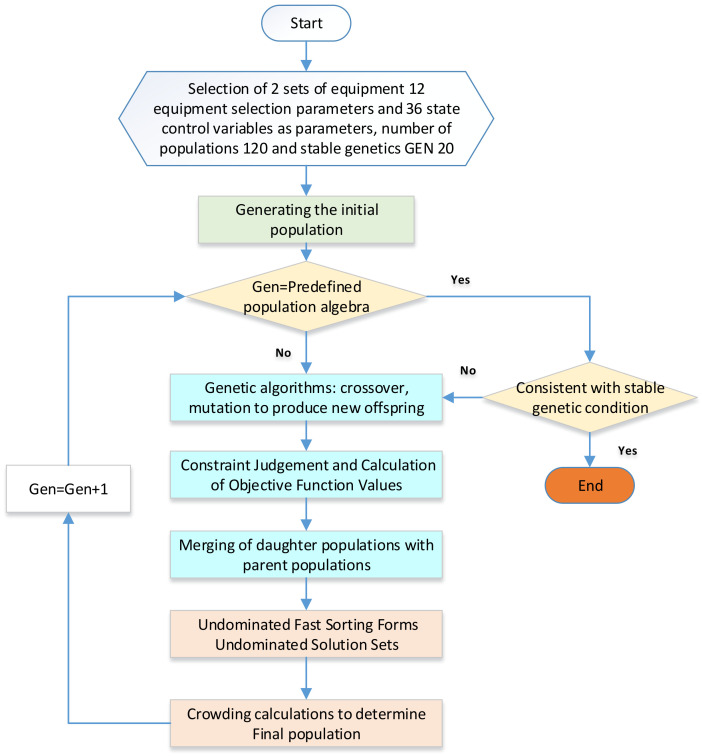
Refrigeration systems for two sets of equipment.

**Figure 8 entropy-25-01386-f008:**
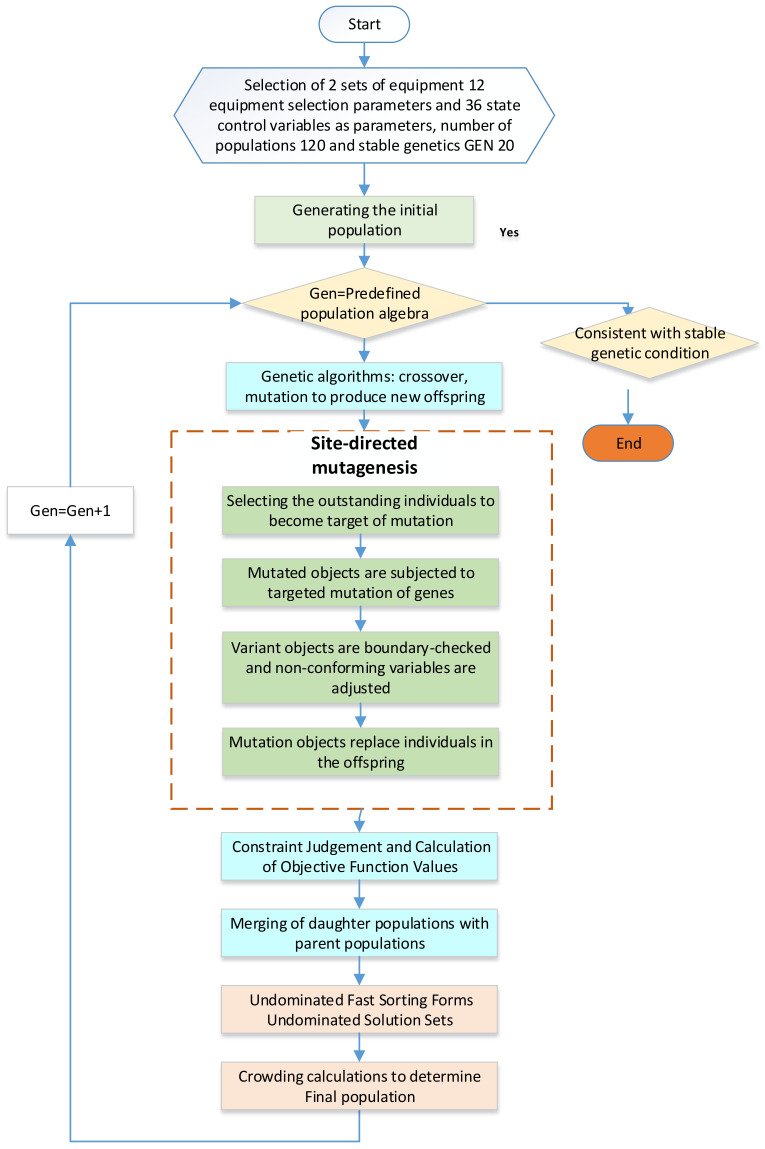
Gene-directed change genetic algorithm.

**Figure 9 entropy-25-01386-f009:**
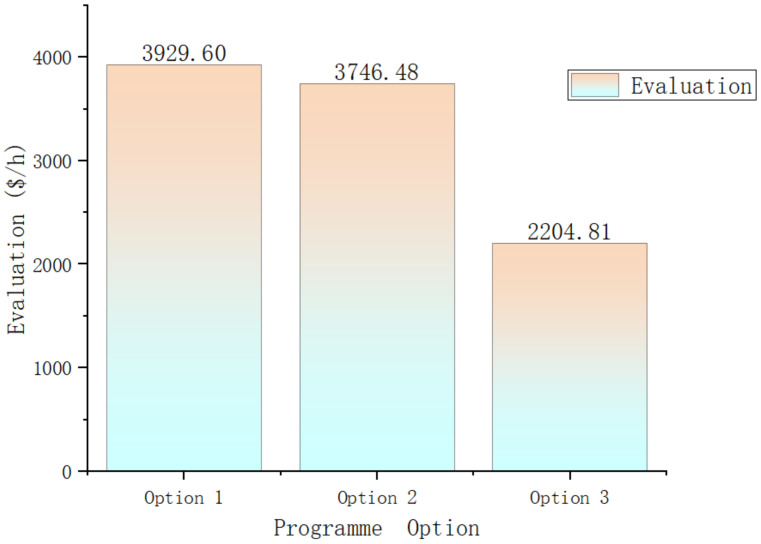
The cooling evaluation coefficient.

**Figure 10 entropy-25-01386-f010:**
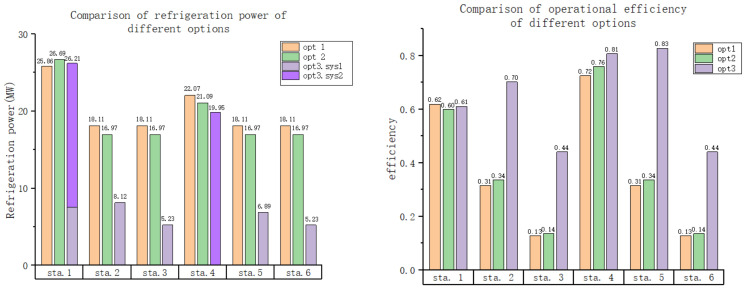
Comparison of the operating efficiencies and amounts of refrigeration power of the different options in various states.

**Figure 11 entropy-25-01386-f011:**
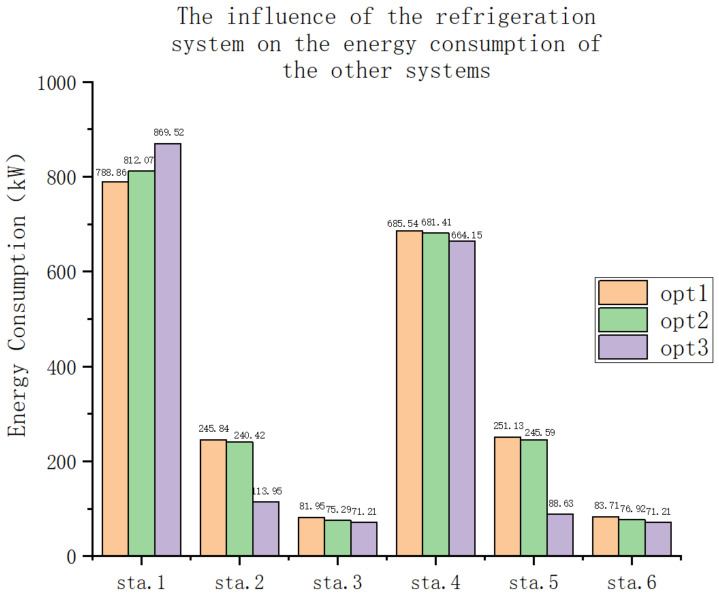
The influence of the refrigeration system on the energy consumption of the other systems.

**Figure 12 entropy-25-01386-f012:**
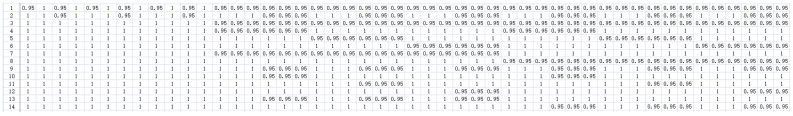
The gene-targeting mutation strategy caption.

**Figure 13 entropy-25-01386-f013:**
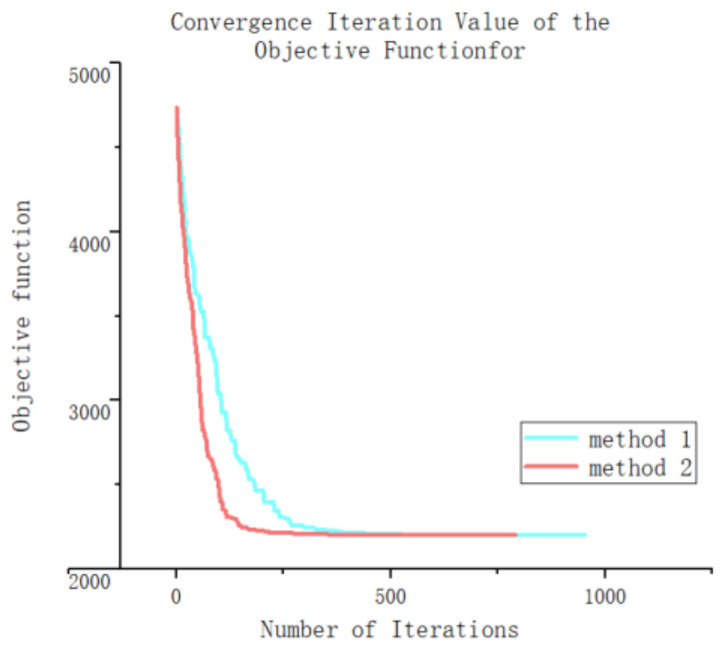
Convergence iteration value of the objective function.

**Figure 14 entropy-25-01386-f014:**
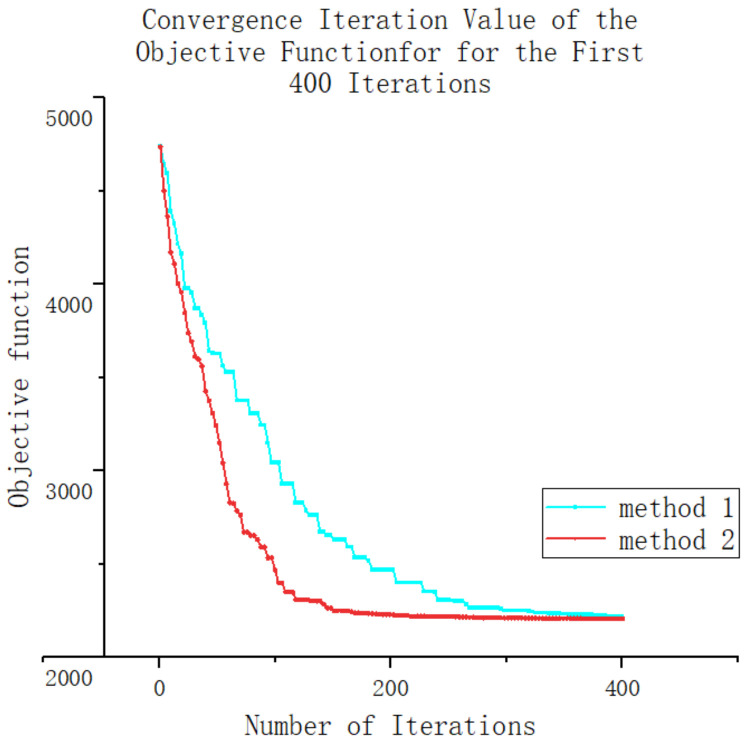
Convergence iteration value of the objective function for the first 400 iterations.

**Table 1 entropy-25-01386-t001:** The key system component selection indexes.

	Descriptions	Realm	Initial Value
1	Centrifugal compressor reaction	0.3–0.8	0.5
2	Centrifugal compressor gauge factor (rated flow kg/s)	200–800	400
3	Compressor reaction	0.3–0.7	0.56
4	Compressor gauge factor (rated flow kg/s)	200–800	400
5	Expander reaction	0.3–0.7	0.56
6	Expander gauge factor (rated flow kg/s)	200–800	400

**Table 2 entropy-25-01386-t002:** The operating parameter indexes of the system.

	Descriptions	Realm	Initial Value
1	Centrifugal compressor input power (kW)	5000–100,000	25,000
2	Compressed air mass flow (kg/s)	100–800	414
3	Cooling water mass flow rate (kg/s)	800–20,000	1000

**Table 3 entropy-25-01386-t003:** The frequency coefficients of the wind tunnel experimental requirements.

	Low Temperature (−70 °C)	Normal Temperature (5 °C)
High speed (100 m/s)	Status 1 (0.1)	Status 4 (0.1)
Medium speed (65 m/s)	Status 2 (0.3)	Status 5 (0.3)
Low speed (40 m/s)	Status 3 (0.1)	Status 6 (0.1)

**Table 4 entropy-25-01386-t004:** Construction costs for the different system options (USD).

	Option 1	Option 2	Option 3	System I	System II
Centrifugal Compressor	1.53 × 10^6^	1.51 × 10^6^		4.79 × 10^5^	1.11 × 10^6^
Compressor	8.90 × 10^5^	8.70 × 10^5^		2.76 × 10^5^	6.41 × 10^5^
Water cooler	8.40 × 10^4^	8.31 × 10^4^		4.17 × 10^4^	6.91 × 10^4^
Re-cooler	1.16 × 10^5^	1.14 × 10^5^		3.62 × 10^4^	8.41 × 10^4^
Rxpander	5.03 × 10^5^	4.93 × 10^5^		1.56 × 10^5^	3.64 × 10^5^
Cooling towers	2.19 × 10^4^	2.15 × 10^4^		6.82 × 10^3^	1.58 × 10^4^
Purification device	5.61 × 10^4^	5.50 × 10^4^		1.75 × 10^4^	4.05 × 10^4^
Total costs	3.21 × 10^6^	3.14 × 10^6^	3.33 × 10^6^	1.01 × 10^6^	2.32 × 10^6^

**Table 5 entropy-25-01386-t005:** Comparison of optimization schemes.

	Method 1 Optimal Value (USD/h)	Method 2 Optimal Value (USD/h)	Method 1 Optimization Algebra	Method 2 Optimizing Algebra
1st	2205.46	2203.83	938	816
2nd	2204.63	2205.20	922	804
3rd	2203.65	2204.96	929	803
4th	2205.14	2205.50	1049	821
5th	2203.71	2205.44	1055	862
6th	2203.87	2203.91	952	788
7th	2204.96	2203.99	1056	804
8th	2203.65	2204.20	920	862
9th	2204.88	2204.03	1060	815
10th	2204.45	2204.73	964	754
Average value			984.4	812.9

## Data Availability

All research data can be accessed via email junjunzh@buaa.edu.cn.

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
