# Peer review of "Optimizing the Cooling System of High-Speed Train Environmental Wind Tunnels Using the Gene-Directed Change Genetic Algorithm"

_entropy, 2023, doi:10.3390/e25101386_

Round 1
Reviewer 1 Report (New Reviewer)
The subject of the paper is interesting, as it uses genetic algorithms when dealing with an optimization problem of industrial interest. There are however two main important points that should be corrected before considering the publication of the paper.
1.- The general explanation of the industrial situation (cooling systems for environmental wind tunnels for high-speed trains) is made in a very confusing way. A reader not familiar with the industrial situation (but perhaps interested in genetic algorithms) can hardly understand what is being analysed and optimized.
2.- From a formal point of view, the paper presents many errors and details that have not been taken care of. These points must be corrected and a list of them follows
1.- Page 4 and 5: use numbers to identify equations
2.- Use numbering of the lines of the paper, so it is possible for the reviewer to refer to specific lines without having to describe the whole paragraph to identify it.
3.- The meaning of each coefficient of the equations should be clearly explained, just after the equation has been written. Units of each coefficient are advised, so the understanding of the equations is easier. If the authors of the manuscripts prefer to do so in an appendix, please refer to the appendix when stating the equations.
4.- The third equation seems to be a matrix multiplication. Please, clarify.
5.- Explanations in section 2 to reach the evaluation factor Gsys (equation 4) are obscure and not clearly related to the formula. Please, improve it.
6.- Section 3: when referring to previous publications of the authors (see line 5 of section 3) please, specify the reference numbers.
7.- Table 1: The initial value of concepts 2, 4 and 6 is outside the realm proposed.
8.- Section 3.1: after figure 5 caption, it is said that optimization is intended for -70ºC. Is this value correct? A quick search in the internet suggests more a -40ºC temperature.
9.- After mentioning figure 8: for the first time, this figure should appear and not after the gene-targeting mutation strategy.
10.- After figure 8, a matrix is mentioned, but no explanation of the matrix is given. Perhaps a reference to figure 12 (if this is the case) could be of use.
11.- Section 4.1: When referring to references in the literature, please indicate the corresponding numbers in the references list.
12.- Figure 9: Option word is repeated. The units of the vertical axis Y (USD/hour), are missing
13.- Table 4: it seems that costs for Option 3 and for System III are missing
14.- Different tables with the same number (Table 3) appear, see pages 15 and 18. The second table is probably number 5. The text of the paragraph that refers to it should be changed accordingly.
15.- Page 19, second line: separation between words is needed in which uses a
16.- Appendix 1: should include units of the variables, which helps to understand their meaning.
English is fine
Author Response
Dear reviewer:
Thank you for reviewing our article. We appreciate the thoroughness with which you assessed it and for providing valuable feedback. Your suggested improvements will undoubtedly enhance our theoretical and simulation research article. Before we respond your comments, we would like to clarify that our article does not present experimental research but rather delves into the theoretical and simulation aspects of our topic. The focus of our investigation is the scheme of the environmental wind tunnel for high-speed trains, presently in development in China. Given its enormous proportions, conducting early verification experiments is challenging. Our initial research will depend on theoretical and simulated verification instead. Concerning your recommendation, we respond as follows:
Remarks 1 The general explanation of the industrial situation (cooling systems for environmental wind tunnels for high-speed trains) is made in a very confusing way. A reader not familiar with the industrial situation (but perhaps interested in genetic algorithms) can hardly understand what is being analysed and optimized
Response 1
Your suggestion regarding our introduction to the wind tunnel cooling system is very helpful. We recognized that our previous explanation was too simplified and have edited it below:
High-speed railways play an increasingly important role in economic transport. As an essential experimental facility for the research and development of high-speed railways, the high-speed-railway environmental wind tunnel has received significant attention. Nowadays, some of the world's famous environmental wind tunnels for trains include the Climatic Wind Tunnel Vienna, the CIRA Icing Wind Tunnel, the McKinley Climatic Laboratory, and the Jules Verne climatic wind tunnel. They have played a significant role in advancing high-speed train technology. One notable example is the Climatic Wind Tunnel Vienna, which boasts an impeccable weather simulation system. China, the world's foremost high-speed rail nation, is yet to possess its own high-speed rail wind tunnels. Since the foundation of The National Innovation Centre of High-Speed Train in 2019, China has been seeking to construct its own environmental wind tunnels. However, the construction and operation of high-speed railway wind tunnels for environmental testing, especially for full-scale models, incurs considerable costs. This is attributed to the vast dimensions and intricacy of the wind tunnels. The wind tunnel for high-speed trains is a large-scale experimental facility consisting of several systems, including the power system, refrigeration system, control system, measurement and data acquisition systems, among others. The refrigeration system, a crucial subsystem for wind tunnel operations, not only provides cooling capacity to simulate a low-temperature environment in the experimental section of the wind tunnel, but also is responsible for dissipating heat from other components in the tunnel. Due to the substantial demand for cooling in the wind tunnel process, the system consumes a significant amount of energy. The refrigeration system is comprised of several components, including suction compressors, turbo-compressors, water coolers, return coolers, and turbo-expanders, resulting in a complex structure. The coordination of these components and the appropriate strategy for their use determine the advantages and disadvantages of the refrigeration system. Therefore, optimizing the design of the refrigeration system can effectively enhance its operating efficiency and reduce costs.
Moreover, a detailed description of the composition and operation of the air compression refrigeration system is provided in section 2 of this article.
The cooling system of a high-speed-train environmental wind tunnel is based on the Brayton inverse cycle and comprises specific components, as shown in Figure 1.The system follows the Brayton cycle in which air is compressed and expanded to generate low-temperature cooling air for the high-speed-train environmental wind tunnel. The main components include the inlet compressor, turbo-compressor, turbo-expander, water cooler, return cooler, and cooling tower.
Figure 1. Air compression refrigeration cycle.
The procedure begins with the compression of external air by the inlet compressor. The compressed air is then dried and purified, resulting in a high-temperature and high-pressure dry, clean gas. After further compression via the turbo-compressor, the pressure and temperature of the air increase. Through the cooling system, the temperature of the compressed air is reduced by passing it through the water cooler and the re-cooler. The cooled, compressed air then enters the turbo-expander, where work is done via expansion to produce low-temperature gases at atmospheric pressure. This cooled air is then sent into the high-speed-train environmental wind tunnel via a pump, serving as the cold source. The power output from the expander can be utilized to drive the co-axial compressor, enabling the recovery of the compressed air’s energy and increasing the energy efficiency of the system.
Remarks 2 From a formal point of view, the paper presents many errors and details that have not been taken care of. These points must be corrected and a list of them follows
2.1.- Page 4 and 5: use numbers to identify equations
Response 2.1
Yes, that's a good suggestion. We have now numbered all the formulas in the article to enhance formula descriptions. Here is the modified version.
This new system of evaluation can be expressed using Equation 1.
|
|
(1) |
Alternatively, equation 2 can be obtained from equation 1:
|
|
(2) |
where represents the economic cost of the experimental cycle, $ /h; andrepresents the economic cost of the component in the experimental cycle, $ /h; and denotes the weighting coefficients for different experimental states in the full-speed cycle of the train experiment. These coefficients are calculated to obtain the experimental cycle weights and can be expressed by using Equation 3.
|
|
(3) |
|
(4) |
|
|
(5) |
Remarks 2.2 Use numbering of the lines of the paper, so it is possible for the reviewer to refer to specific lines without having to describe the whole paragraph to identify it.
Response 2.2
This is a very good suggestion and we couldn't agree more, but since the templates provided by entropy magazine have the line numbering feature disabled in our system we are unable to implement it. We apologize for any inconvenience caused and kindly request that you provide feedback on the issue by noting down the page number or annotating the article. We appreciate your cooperation and will be happy to communicate with you on the matter.
Remarks 2.3 The meaning of each coefficient of the equations should be clearly explained, just after the equation has been written. Units of each coefficient are advised, so the understanding of the equations is easier. If the authors of the manuscripts prefer to do so in an appendix, please refer to the appendix when stating the equations.
Response 2.3
We apologize for the unclear description of our evaluation system. Thus, we have rewritten each formula's description to enhance comprehension for our readers.
To address these related problems, a detailed energy flow analysis of the complete system of the high-speed train environmental wind tunnel should be conducted first to clarify the energy flow coupling between the refrigeration system and the other systems in a high-speed-train environmental wind tunnel. Subsequently, we analyze and quantify the exergoeconomic cost of the refrigeration system in relation to the other systems present in the wind tunnel VOs, $/h. VOs together with the previous Fsys ,comprises the final evaluation factor for the refrigeration system, known as Gsys ,$/h. Equation 4 demonstrates how Gsys can be rewritten to determine its final equation.
|
(4) |
Here, represents the external impact values of the systems,$/h.
Remarks 2.4 The third equation seems to be a matrix multiplication. Please, clarify.
Response 2.4
I apologise for the previous erroneous equation and have now provided the corrected version. This equation elucidates the method for determining the weight coefficient of the system usage frequency T.
Tk denotes the weighting coefficients for different experimental states in the full-speed cycle of the train experiment. These coefficients are calculated to obtain the experimental cycle weights and can be expressed by using Equation 3.
|
|
(3) |
Remarks 2.5 Explanations in section 2 to reach the evaluation factor Gsys (equation 4) are obscure and not clearly related to the formula. Please, improve it.
Response 2.5
We have improved the description of Equation 2.5, here is the revised version and your feedback is appreciated.
To address these related problems, a detailed energy flow analysis of the complete system of the high-speed train environmental wind tunnel should be conducted first to clarify the energy flow coupling between the refrigeration system and the other systems in a high-speed-train environmental wind tunnel. Subsequently, we analyze and quantify the exergoeconomic cost of the refrigeration system in relation to the other systems present in the wind tunnel VOs, $/h. VOs together with the previous Fsys ,comprises the final evaluation factor for the refrigeration system, known as Gsys ,$/h. Equation 4 demonstrates how Gsys can be rewritten to determine its final equation.
|
(4) |
Here, represents the external impact values of the systems,$/h.
Remarks 2.6 Section 3: when referring to previous publications of the authors (see line 5 of section 3) please, specify the reference numbers.
Response 2.6
We have changed the presentation of the article here。
The modelling and sensitivity analyses of design parameters for air compression refrigeration systems have been discussed in a published article[26]. Optimization function variables can be categorized into two main groups: system component selection indexes and operating parameter indexes.
- Zhuang J, Liu M, Wu H, Wang J. Designing an Environmental Wind Tunnel Cooling System for High-Speed Trains with Air Compression Cooling and a Sensitivity Analysis of Design Parameters[J]. Entropy, 2023, 25(9): 1312.
Remarks 2.7 Table 1: The initial value of concepts 2, 4 and 6 is outside the realm proposed
Response 2.7
The data in this table was incorrect when we modified it, but this is the correct table. I'm really sorry.
Table 1. The key system component selection indexes.
|
Descriptions |
Realm |
Initial Value |
|
|
1 |
Centrifugal compressor reaction |
0.3–0.8 |
0.5 |
|
2 |
Centrifugal compressor gauge factor (rated flow kg/s) |
200-800 |
400 |
|
3 |
Compressor reaction |
0.3–0.7 |
0.56 |
|
4 |
Compressor gauge factor (rated flow kg/s) |
200-800 |
400 |
|
5 |
Expander reaction |
0.3–0.7 |
0.56 |
|
6 |
Expander gauge factor (rated flow kg/s) |
200-800 |
400 |
Remarks 2.8 Section 3.1: after figure 5 caption, it is said that optimization is intended for -70ºC. Is this value correct? A quick search in the internet suggests more a -40ºC temperature.
Response 2.8
It's -70°C here. Our newly optimised train wind tunnel is designed for use in China,where temperatures can reach even colder levels. In fact, the lowest temperature recorded in Chinese cities in 2021 was -48.9°C, and the temperature during actual train operation in the field could be as low as -60°C. Due to the inability of the existing -40°C wind tunnel to simulate these extreme conditions, we believe that the lowest simulated temperature should be set to -70°C.
Remark 2.9. After mentioning figure 8: for the first time, this figure should appear and not after the gene-targeting mutation strategy.
Response 2.9
You are right, we have adjusted the position of this image and the revised version is shown below.
…However, during the initial phases, the disorderly nature of evolution leads to anoverall low system efficiency. By using fixed-point gene mutation technology, we can effectively address this issue. The specific operational steps are as follow inFigure 8:
Figure 8. Gene-directed-change genetic algorithm.
The gene-targeting mutation strategy is outlined below:
- Reduce the dimensional design parameters of all components (excluding the compressor and turbine inversions) and system control variables by 5%.
…
Remark 2.10. After figure 8, a matrix is mentioned, but no explanation of the matrix is given. Perhaps a reference to figure 12 (if this is the case) could be of use.
Response 2.10
Yes,figure 12 displays the matrix. The matrix is not presented here as the value of each matrix can be tailored based on the actual test outcome. Therefore, we do not provide a written version of this matrix in this section and it can be found in Section 4.2.
Remark 2.11. Section 4.1: When referring to references in the literature, please indicate the corresponding numbers in the references list.
Response 2.11
Sorry, our wording may have caused confusion. We actually have no cited literature here, or we refer to a Chinese railway experimental standard number is TB/1675 - 2001 "Railway Passenger Car Air Conditioning Test Methods" in the train low temperature experiment to determine the provisions. We also consulted with some experimental personnel, who found this setting appropriate. We've made the change here to prevent misinterpretation,
According to industry standards and engineering expertise, the experimental requirements for high-speed train testing generally follow a 1:3:1 time ratio of high speed/conventional speed/low speed
Remark 2.12. Figure 9: Option word is repeated. The units of the vertical axis Y (USD/hour), are missing
Response 2.12
Yes, sorry, we've changed it.
Remark 2.13. Table 4: it seems that costs for Option 3 and for System III are missing
Response 2.13
There is no missing information as the original text lacked context. I apologise for any confusion caused. In fact Option 3 comprises two systems, System 1 and System 2. Therefore, the table in Option 3 is currently empty. And Option 3 only has two systems: system 1 and system 2. There is no system 3.
Remark 2.14. Different tables with the same number (Table 3) appear, see pages 15 and 18. The second table is probably number 5. The text of the paragraph that refers to it should be changed accordingly.
Response 2.14
Yes you are right. I'm really sorry to say that once again we made a typographical error. Now we have the article corrected.
The optimal solutions and the number of optimization generations for each dataset are presented in Table 5.
…..
Table 5. Comparison of optimization schemes.
|
Method 1 Optimal Value ($/h) |
Method 2 Optimal Value ($/h) |
Method 1 Optimization Algebra |
Method 2 Optimizing Algebra |
|
|
1st |
2205.46 |
2203.83 |
938 |
816 |
|
2nd |
2204.63 |
2205.20 |
922 |
804 |
|
3rd |
2203.65 |
2204.96 |
929 |
803 |
|
4th |
2205.14 |
2205.50 |
1049 |
821 |
|
5th |
2203.71 |
2205.44 |
1055 |
862 |
|
6th |
2203.87 |
2203.91 |
952 |
788 |
|
7th |
2204.96 |
2203.99 |
1056 |
804 |
|
8th |
2203.65 |
2204.20 |
920 |
862 |
|
9th |
2204.88 |
2204.03 |
1060 |
815 |
|
10th |
2204.45 |
2204.73 |
964 |
754 |
|
Average value |
984.4 |
812.9 |
||
Remark 2.15. Page 19, second line: separation between words is needed in which uses a
Response 2.15
You are right. Thanks again for your help, and we have thoroughly reviewed the paper, identified and corrected the errors. However, if there are any remaining typos, please inform us and we will gladly make corrections.
Taking the sixth calculation of both methods as an example, this section analyzes the advantages and disadvantages of the genetic optimization algorithm, which uses a gene fixation mutation, in detail.
Remark 2.16. Appendix 1: should include units of the variables, which helps to understand their meaning.
Response 2.16
We appreciate the suggestion and have included the units for the variables.
|
symbols list |
subscript variable symbol: |
||
|
k |
comment |
||
|
C |
exergoeconomic cost ($/h) |
sys |
system |
|
E |
exergoeconomic cost of the systems change. ($/h) |
cyc |
refrigerating system |
|
F |
exergoeconomic cost with frequency coefficient ($/h) |
h |
high temperature |
|
G |
exergoeconomic cost with an Experimental Cycle ($/h) |
c |
normal temperature |
|
T |
frequency of environmental wind tunnel |
hs |
high speed |
|
Z |
construction cost of this system ($/h) |
ms |
medium speed |
|
q |
cooling capacity |
ls |
low speed |
|
efficiency |
os |
the external impact |
|
|
expansion ratio |
shi |
actual |
|
|
mass flow (kg/s) |
xu |
requirement |
|
We hope our response meets your expectations. Yours advice are very helpful, thank you for your advice again. Please do not hesitate to let us know if you have any further suggestions in the future.
Kind Regards,
Yours sincerely,
Junjun Zhuang

Reviewer 2 Report (New Reviewer)
In this work, has been focused on the refrigeration system of high-speed train environmental wind tunnel; a new system of evaluation for a refrigeration system that is based on the system’s energy and economic performance is proposed. The research is attractive for engineering applications. The paper is quite of good quality, but some requests and suggestions have been provided to increase the quality of the work.
The introduction should be improved by adding also other methodology for the evaluation of the refrigeration as the CFD model. For example, you can add:
- Broniszewski, J.; Piechna, J.R. Fluid-Structure Interaction Analysis of a Competitive Car during Brake-in-Turn Manoeuvre. Energies 2022, 15, 2917.
- Cravero, C.; Marsano, D. Flow and Thermal Analysis of a Racing Car Braking System. Energies 2022, 15, 2934.
These paper reports some CFD models for the fluid dynamic analysis concerning the cooling of a component like the brake for automotive applications, but similar models can be considered for you.
Moreover I suggest to consider the uncertainty on the inputs. So as example these works report the state of the art of the surrogate models:
- Xia, L., Zou, Z. J., Wang, Z. H., Zou, L., & Gao, H. Surrogate model based uncertainty quantification of CFD simulations of the viscous flow around a ship advancing in shallow water. Ocean Engineering 2021, 234, 109206
- Cravero, C; De Domenico, D; Marsano, D. “Uncertainty Quantification Analysis of Exhaust Gas Plume in a Crosswind”. Energies, 2023, Vol. 16, Issue 8, p. 3549.
The ergonomic analysis with experimental cycle should be improved with more details on the experimental instrumentation with also some figure.
The optimization model has been well described with many diagrams and equations, however you should clarify better the software adopted.
The results section is very complete; however some figures are too difficult to read, please improve them.
The conclusions are well supported by the results and it is clear your contribution.
Minor typos
Author Response
Dear reviewer:
Thank you for reviewing our article. Your endorsement is greatly appreciated. We have carefully considered your comments and made the necessary changes to the essay accordingly.
Remarks 1. The paper is quite of good quality, but some requests and suggestions have been provided to increase the quality of the work.
The introduction should be improved by adding also other methodology for the evaluation of the refrigeration as the CFD model.
Moreover I suggest to consider the uncertainty on the inputs.
Response 1
Your advice was especially helpful, and we found the references you provided to be very inspiring. We would be delighted to include your suggestions in our next article. Here's the revised part of our article:
The environmental wind tunnel for high-speed trains must be tested under various experimental conditions, and a system model should be developed to analyze the operational mode. An in-depth analysis of the refrigeration system and wind tunnel operation under different operating conditions is crucial. Broniszewski [13] researched the impact of different fluid-structures on wind tunnel resistance, and Cravero [14] further examined the correlation between fluid flow and heat. Due to the large number of components involved in the refrigeration system, uncertainty in the inputs should be considered when selecting parameters. Xia [15] examined the uncertainty related to the flow field input, while Cravero [16] carried out further analysis of the factors contributing to the uncertainty under varying flow conditions. Despite sensitivity analysis allowing for the removal of low sensitivity design parameters in the refrigeration system, there are still numerous design parameters that necessitate optimisation. Due to the large number of variables involved, traditional numerical solution methods are not feasible; thus a genetic algorithm is chosen to optimally solve the cooling system. The genetic algorithm was first proposed by Professor Holland[17]in 1975 and has become a mainstream intelligent algorithm that is widely applied to thermodynamic problems[18–21]. However, this algorithm still has some shortcomings. For example, when the objective function has multiple peaks, the genetic algorithm tends to obtain local optimal solutions, leading to “genetic drift”[22]. Ajimakin et al.[23]proposed the “Genes Evolutionary Algorithm”, a binary algorithm that uses a binary function search space to avoid genetic drift by optimizing the fitness function. To improve the speed and accuracy of the solution, Giovanni et al.[24] refined the process of using genetic algorithms to solve the distributed and flexible job-shop scheduling problem by introducing a new local search-based operator in addition to traditional crossover and mutation operators. Jia Xu et al.[25] introduced the gene sequence comparison method from biology to improve the crossover operation, resulting in a faster solution speed and more stable results.
- Broniszewski, J.; Piechna, J.R. Fluid-Structure Interaction Analysis of a Competitive Car during Brake-in-Turn Manoeuvre. Energies 2022, 15, 2917.B
- Cravero, C.; Marsano, D. Flow and Thermal Analysis of a Racing Car Braking System. Energies 2022, 15, 2934.
- Xia, L., Zou, Z. J., Wang, Z. H., Zou, L., & Gao, H. Surrogate model based uncertainty quantification of CFD simulations of the viscous flow around a ship advancing in shallow water. Ocean Engineering 2021, 234, 109206
- Cravero, C; De Domenico, D; Marsano, D. “Uncertainty Quantification Analysis of Exhaust Gas Plume in a Crosswind”. Energies, 2023, Vol. 16, Issue 8, p. 3549.
Remarks 2 The ergonomic analysis with experimental cycle should be improved with more details on the experimental instrumentation with also some figure.
Response 2
We have taken your suggestion very seriously and we have modified the description of the parameters in the hope that readers will better understand the meaning of the article.
…
Therefore, the weight coefficient of the system usage frequency can be increased based on the economic success to describe the frequency of environmental wind tunnel usage in different states during the complete cycle of a high-speed-train environmental experiment. In this way, a new system of evaluating environmental wind tunnels based on the experimental cycle of a high-speed railway train can be formed. This new system of evaluation can be expressed using Equation 1.
|
|
(1) |
Alternatively, equation 2 can be obtained from equation 1:
|
|
(2) |
where represents the economic cost of the experimental cycle, $/h;represents the economic cost of the component in the experimental cycle, $/h; and denotes the weighting coefficients for different experimental states in the full-speed cycle of the train experiment. These coefficients are calculated to obtain the experimental cycle weights and can be expressed by using Equation 3.
|
|
(3) |
The impact of the refrigeration system’s energy consumption on the other systems in the wind tunnel can be addressed by conducting an exergy analysis on the entire experimental system as a whole. While this method accurately analyzes the influence of changes in the refrigeration system on other systems, it also significantly increases the number of analysis systems, thereby increasing the analysis cost. Moreover, it may fail to highlight the energy flow characteristics of the refrigeration system due to excessive redundant data, thus complicating the discovery of the refrigeration system’s energy flow coupling law. To address these related problems, a detailed energy flow analysis of the complete system of the high-speed train environmental wind tunnel should be conducted first to clarify the energy flow coupling between the refrigeration system and the other systems in a high-speed-train environmental wind tunnel. Subsequently, we analyze and quantify the exergoeconomic cost of the refrigeration system in relation to the other systems present in the wind tunnel VOs, $/h. VOs together with the previous Fsys ,comprises the final evaluation factor for the refrigeration system, known as Gsys ,$/h. Equation 4 demonstrates how Gsys can be rewritten to determine its final equation.
|
|
(4) |
Here, represents the external impact values of the systems,$/h.
This evaluation index enables a more effective comparison of the advantages and disadvantages of different environmental cooling systems for high-speed trains. Consequently, it serves as the main objective function for the subsequent optimization of environmental cooling.
Remarks 3: The optimization model has been well described with many diagrams and equations, however you should clarify better the software adopted.
Response 3
This suggestion was carefully considered. The software used and simulation environment are described later in the optimisation as they are crucial and essential. Therefore, we did not introduce the software used during earlier model optimisation.
In this paper, the process was programmed using MATLAB 2018b, based on the improved algorithm mentioned earlier, to optimize and analyze the method of selection and mode of operation of the high-speed-train environmental wind tunnel. To verify the effectiveness and feasibility of the second method and compare the running speeds of the two algorithms, each algorithm was independently executed 10 times on a computer system, using an Intel(R) Core(TM) i7-3770K CPU @ 3.50GHz and 16GB RAM. Simulink’s model was used to run the wind tunnel refrigeration models.
Remarks 4:The results section is very complete; however some figures are too difficult to read, please improve them.
Response 4
When you say it is too hard to read you are probably talking about the matrix of figure12. We have thoroughly considered this issue, but unfortunately have not found a satisfactory solution. If you have any suggestions, please do share them with us.
We hope our response meets your expectations. Yours advice are very helpful, thank you for your advice again. Please do not hesitate to let us know if you have any further suggestions in the future.
Kind Regards,
Yours sincerely,
Junjun Zhuang

Reviewer 3 Report (New Reviewer)
This study proposes the use of an exergoeconomic evaluation coefficient combined with experimental cycles to establish a wind tunnel refrigeration system. By employing the newly developed evaluation coefficient as an objective function, we utilize the adaptive value-sharing congestion genetic algorithm to optimize the wind tunnel for high-speed trains. This is an interesting topic. The writing of the paper is relatively good; the method is introduced in detail, and the results are well-documented. However, there are some significant concerns that the authors should address seriously. These concerns could be addressed to make the paper acceptable. The main comments can be found below.
(1) The abstract would benefit from a quantitative analysis to enhance the quality of the new method.
(2) In the introduction section, the description of "Zhou et al. [3–7] analyzed and optimized reversible and irreversible reheat air refrigeration cycles under constant" is not professional. The way the references are lumped together is not appropriate.
(3) It is recommended to reorganize the structure of sections two and three. Subsections could make the manuscript clearer and more convincing. Additionally, section 3.1 could be arranged a little bit earlier.
(4) Figures 2 and 3, which depict the turbine and heat exchanger modeling flowcharts, respectively, may need to provide reliable information for the entire content.
(5) The authors need to provide the equation sequence and the references that prove the correct source.
(6) For the cooling evaluation coefficient per unit of time, the authors need to provide sufficient background information.
(7) In Table 4, "Construction costs for the different system options($)", some information seems to be missing for Option 3.
(8) The caption for Figure 12, "The gene-targeting mutation strategy," could be presented in a table format. Please consider redrawing the table or creating an interesting picture.
(9) For Table 3, "Comparison of optimization schemes," the authors need to provide the units for the values.
(10) The reviewer cannot fully understand the meaning of the vertical axis unit in relation to the convergence iteration value of the objective function.
Author Response
Dear reviewer:
Thank you for acknowledging our dissertation. Your feedback on our paper was invaluable in improving its quality. We appreciate your assistance in this matter.
We have carefully considered your comments and made the necessary changes to the essay accordingly.
Remarks 1 The abstract would benefit from a quantitative analysis to enhance the quality of the new method.
Response 1
You are quite right, and we have added a quantitative analysis to the abstract of our article.
Furthermore, we compare the advantages and disadvantages of different optimization schemes. Traditional optimization methods prove inefficient due to the system’s numerous variables and the presence of multiple peaks in the objective function. Inspired by the biogenetic breeding method, we introduce an optimization approach based on a specific gene mutation. This innovative method significantly reduces optimization time and improves efficiency by approximately 17%..
Remarks 2 In the introduction section, the description of "Zhou et al. [3–7] analyzed and optimized reversible and irreversible reheat air refrigeration cycles under constant" is not professional. The way the references are lumped together is not appropriate.
Response 2
You are right, it does seem a bit irregular to cite it that way. The reason we did that at the time was because all of his articles were on optimisation of air compression refrigeration cycles, and these articles are very much linked, they are all on different focuses of refrigeration systems. Now we do feel that this is a bit of a standard, and we have revised this part of the article.
Chen et al.[2] further investigated the optimal distribution of thermal conductivity in the actual reheat air cooling cycle and obtained the optimal cooling rate and coefficient.In addition to traditional objectives such as the refrigeration coefficient, Zhou et al. analyzed and optimized reversible [3] and irreversible [4] reheat air refrigeration cycles under constant [5] and variable temperature [6] heat source conditions using the refrigeration rate density (the refrigeration rate divided by the maximum specific volume) as the thermodynamic optimization objective, and they then applied this methodology to optimize the cooling system for solar cells[7].While the energy analysis method effectively describes changes in energy consumption, it fails to evaluate the system’s work capacity.
Remarks 3 It is recommended to reorganize the structure of sections two and three. Subsections could make the manuscript clearer and more convincing. Additionally, section 3.1 could be arranged a little bit earlier.
Response 3
Thank you very much for this suggestion, which we have indeed considered for a long time. Initially, we intended to introduce section 3.1 at the beginning. However, during system optimization, we discovered that none of the existing refrigeration system evaluation methods could serve as the objective function for our optimization. Therefore, we needed to re-establish an objective function, and this led to the creation of Section 2, in which we established a new cooling system evaluation system. However, writing in this manner creates a division between the objective function and the optimization parameters. As a result, we have decided to introduce the evaluation system initially and then describe the method for optimizing the system.
Remarks 4 Figures 2 and 3, which depict the turbine and heat exchanger modeling flowcharts, respectively, may need to provide reliable information for the entire content.
Response 4
We fully support your suggestion which indeed concerns us greatly. We would like to provide a more detailed description of the model, but doing so could detract from the focus of our paper. Instead, we will emphasize our new evaluation function and genetic algorithm in this thesis. In our previous article, “Designing an Environmental Wind Tunnel Cooling System for High-Speed Trains with Air Compression Cooling and a Sensitivity Analysis of Design Parameters”, we provided an extensive account of the methodology for the model. We also performed a sensitivity analysis of the crucial parameters, ultimately identifying the variables that warrant optimization.
Remarks 5 The authors need to provide the equation sequence and the references that prove the correct source.
Response 5
We appreciate your valuable suggestion and have made necessary revisions. The equations are renumbered to ensure their accurate citation and we have also updated the reference citation. Please find the revised version complying with your requirements.
…
Therefore, the weight coefficient of the system usage frequency can be increased based on the economic success to describe the frequency of environmental wind tunnel usage in different states during the complete cycle of a high-speed-train environmental experiment. In this way, a new system of evaluating environmental wind tunnels based on the experimental cycle of a high-speed railway train can be formed. This new system of evaluation can be expressed using Equation 1.
|
|
(1) |
Alternatively, equation 2 can be obtained from equation 1:
|
|
(2) |
where represents the economic cost of the experimental cycle, $/h;represents the economic cost of the component in the experimental cycle, $/h; and denotes the weighting coefficients for different experimental states in the full-speed cycle of the train experiment. These coefficients are calculated to obtain the experimental cycle weights and can be expressed by using Equation 3.
|
|
(3) |
The impact of the refrigeration system’s energy consumption on the other systems in the wind tunnel can be addressed by conducting an exergy analysis on the entire experimental system as a whole. While this method accurately analyzes the influence of changes in the refrigeration system on other systems, it also significantly increases the number of analysis systems, thereby increasing the analysis cost. Moreover, it may fail to highlight the energy flow characteristics of the refrigeration system due to excessive redundant data, thus complicating the discovery of the refrigeration system’s energy flow coupling law. To address these related problems, a detailed energy flow analysis of the complete system of the high-speed train environmental wind tunnel should be conducted first to clarify the energy flow coupling between the refrigeration system and the other systems in a high-speed-train environmental wind tunnel. Subsequently, we analyze and quantify the exergoeconomic cost of the refrigeration system in relation to the other systems present in the wind tunnel VOs, $/h. VOs together with the previous Fsys ,comprises the final evaluation factor for the refrigeration system, known as Gsys ,$/h. Equation 4 demonstrates how Gsys can be rewritten to determine its final equation.
|
|
(4) |
Here, represents the external impact values of the systems,$/h.
This evaluation index enables a more effective comparison of the advantages and disadvantages of different environmental cooling systems for high-speed trains. Consequently, it serves as the main objective function for the subsequent optimization of environmental cooling.
…
The modelling and sensitivity analyses of design parameters for air compression refrigeration systems have been discussed in a published article[26]. Optimization function variables can be categorized into two main groups: system component selection indexes and operating parameter indexes.
- Zhuang J, Liu M, Wu H, Wang J. Designing an Environmental Wind Tunnel Cooling System for High-Speed Trains with Air Compression Cooling and a Sensitivity Analysis of Design Parameters[J]. Entropy, 2023, 25(9): 1312.
Remarks 6 For the cooling evaluation coefficient per unit of time, the authors need to provide sufficient background information.
Response 6
You are right. It is necessary to provide further details. And you might refer to Section 2 for a thorough explanation. Additionally, we have included a definition of the cooling evaluation coefficient and the Exergoeconomic analysis in the introduction. This can be seen below:
….
The exergoeconomic analysis method takes into account both the thermal performance and the construction and maintenance costs of asystem while maximizing its thermodynamic performance. Tyagiet al. [2]studied and optimized a reheat irreversible refrigeration machine using the economics of energy, targeting the refrigeration rate per unit cost. Razmi [11] undertook an Exergoeconomic analysis of the refrigeration system and optimised the refrigeration system with the unit cost of refrigeration as one of the optimisation objectivesCurrently, the exergoeconomic analysis method is widely used in various thermal systems, such as solar-assisted power generation systems, hydropower cogeneration units, and gas–steam combined cycle units.
[11] Razmi A R, Arabkoohsar A, Nami H. Thermoeconomic analysis and multi-objective optimization of a novel hybrid absorption/recompression refrigeration system[J]. Energy, 2020, 210: 118559.
In fact the cooling evaluation coefficien is obtained from Exergoeconomic analysis.
The goal of this exergy based cost assessment is to specify the cost development practices, as well as the calculation of the cost per exergy unit of the product, using for better economic performance of the system To realize the unit cost of exergy streams,cost-balancing equations along with the needed auxiliary equations are adopted to the cycle equipment. The cost balance equation for the system components in exergy costing process is defined by:
|
here, |
|
where c presents the cost per unit of each exergy stream. Also, CW,k and CQ,k are respectively the cost rates related to the generated power and received thermal energy by the component.
Remarks 7 In Table 4, "Construction costs for the different system options($)", some information seems to be missing for Option 3.
Response 7
There is no missing information as the original text lacked context. I apologise for any confusion caused. In fact Option 3 comprises two systems, System 1 and System 2. Therefore, the table in Option 3 is currently empty.
Remarks 8 The caption for Figure 12, "The gene-targeting mutation strategy," could be presented in a table format. Please consider redrawing the table or creating an interesting picture.
Response 8
We have reviewed your suggested approach extensively and executed various methods. The matrix has been presented in a table, as displayed below. So we have no choice but to present it in this visual form. Kindly share any valuable proposals you may have. Thank you in advance.
|
1 |
0.95 |
1 |
0.95 |
1 |
0.95 |
1 |
0.95 |
1 |
0.95 |
1 |
0.95 |
1 |
0.95 |
0.95 |
0.95 |
0.95 |
0.95 |
0.95 |
0.95 |
0.95 |
0.95 |
0.95 |
0.95 |
0.95 |
0.95 |
0.95 |
0.95 |
0.95 |
0.95 |
0.95 |
0.95 |
0.95 |
0.95 |
0.95 |
0.95 |
0.95 |
0.95 |
0.95 |
0.95 |
0.95 |
0.95 |
0.95 |
0.95 |
0.95 |
0.95 |
0.95 |
0.95 |
0.95 |
|
2 |
1 |
1 |
0.95 |
1 |
1 |
1 |
0.95 |
1 |
1 |
1 |
0.95 |
1 |
1 |
1 |
1 |
0.95 |
0.95 |
0.95 |
1 |
1 |
1 |
0.95 |
0.95 |
0.95 |
1 |
1 |
1 |
0.95 |
0.95 |
0.95 |
1 |
1 |
1 |
0.95 |
0.95 |
0.95 |
1 |
1 |
1 |
0.95 |
0.95 |
0.95 |
1 |
1 |
1 |
0.95 |
0.95 |
0.95 |
|
3 |
1 |
1 |
1 |
1 |
1 |
1 |
1 |
1 |
1 |
1 |
1 |
1 |
0.95 |
0.95 |
0.95 |
0.95 |
0.95 |
0.95 |
0.95 |
0.95 |
0.95 |
0.95 |
0.95 |
0.95 |
0.95 |
0.95 |
0.95 |
0.95 |
0.95 |
0.95 |
0.95 |
0.95 |
0.95 |
0.95 |
0.95 |
0.95 |
0.95 |
0.95 |
0.95 |
0.95 |
0.95 |
0.95 |
0.95 |
0.95 |
0.95 |
0.95 |
0.95 |
0.95 |
|
4 |
1 |
1 |
1 |
1 |
1 |
1 |
1 |
1 |
1 |
1 |
1 |
1 |
0.95 |
0.95 |
0.95 |
0.95 |
0.95 |
0.95 |
1 |
1 |
1 |
1 |
1 |
1 |
1 |
1 |
1 |
1 |
1 |
1 |
0.95 |
0.95 |
0.95 |
0.95 |
0.95 |
0.95 |
1 |
1 |
1 |
1 |
1 |
1 |
1 |
1 |
1 |
1 |
1 |
1 |
|
5 |
1 |
1 |
1 |
1 |
1 |
1 |
1 |
1 |
1 |
1 |
1 |
1 |
1 |
1 |
1 |
1 |
1 |
1 |
0.95 |
0.95 |
0.95 |
0.95 |
0.95 |
0.95 |
1 |
1 |
1 |
1 |
1 |
1 |
1 |
1 |
1 |
1 |
1 |
1 |
0.95 |
0.95 |
0.95 |
0.95 |
0.95 |
0.95 |
1 |
1 |
1 |
1 |
1 |
1 |
|
6 |
1 |
1 |
1 |
1 |
1 |
1 |
1 |
1 |
1 |
1 |
1 |
1 |
1 |
1 |
1 |
1 |
1 |
1 |
1 |
1 |
1 |
1 |
1 |
1 |
0.95 |
0.95 |
0.95 |
0.95 |
0.95 |
0.95 |
1 |
1 |
1 |
1 |
1 |
1 |
1 |
1 |
1 |
1 |
1 |
1 |
0.95 |
0.95 |
0.95 |
0.95 |
0.95 |
0.95 |
|
7 |
1 |
1 |
1 |
1 |
1 |
1 |
1 |
1 |
1 |
1 |
1 |
1 |
0.95 |
0.95 |
0.95 |
0.95 |
0.95 |
0.95 |
0.95 |
0.95 |
0.95 |
0.95 |
0.95 |
0.95 |
0.95 |
0.95 |
0.95 |
0.95 |
0.95 |
0.95 |
1 |
1 |
1 |
1 |
1 |
1 |
1 |
1 |
1 |
1 |
1 |
1 |
1 |
1 |
1 |
1 |
1 |
1 |
|
8 |
1 |
1 |
1 |
1 |
1 |
1 |
1 |
1 |
1 |
1 |
1 |
1 |
1 |
1 |
1 |
1 |
1 |
1 |
1 |
1 |
1 |
1 |
1 |
1 |
1 |
1 |
1 |
1 |
1 |
1 |
0.95 |
0.95 |
0.95 |
0.95 |
0.95 |
0.95 |
0.95 |
0.95 |
0.95 |
0.95 |
0.95 |
0.95 |
0.95 |
0.95 |
0.95 |
0.95 |
0.95 |
0.95 |
|
9 |
1 |
1 |
1 |
1 |
1 |
1 |
1 |
1 |
1 |
1 |
1 |
1 |
1 |
1 |
1 |
0.95 |
0.95 |
0.95 |
1 |
1 |
1 |
0.95 |
0.95 |
0.95 |
1 |
1 |
1 |
0.95 |
0.95 |
0.95 |
1 |
1 |
1 |
0.95 |
0.95 |
0.95 |
1 |
1 |
1 |
0.95 |
0.95 |
0.95 |
1 |
1 |
1 |
0.95 |
0.95 |
0.95 |
|
10 |
1 |
1 |
1 |
1 |
1 |
1 |
1 |
1 |
1 |
1 |
1 |
1 |
1 |
1 |
1 |
0.95 |
0.95 |
0.95 |
1 |
1 |
1 |
1 |
1 |
1 |
1 |
1 |
1 |
1 |
1 |
1 |
1 |
1 |
1 |
0.95 |
0.95 |
0.95 |
1 |
1 |
1 |
1 |
1 |
1 |
1 |
1 |
1 |
1 |
1 |
1 |
|
11 |
1 |
1 |
1 |
1 |
1 |
1 |
1 |
1 |
1 |
1 |
1 |
1 |
1 |
1 |
1 |
1 |
1 |
1 |
1 |
1 |
1 |
0.95 |
0.95 |
0.95 |
1 |
1 |
1 |
1 |
1 |
1 |
1 |
1 |
1 |
1 |
1 |
1 |
1 |
1 |
1 |
0.95 |
0.95 |
0.95 |
1 |
1 |
1 |
1 |
1 |
1 |
|
12 |
1 |
1 |
1 |
1 |
1 |
1 |
1 |
1 |
1 |
1 |
1 |
1 |
1 |
1 |
1 |
1 |
1 |
1 |
1 |
1 |
1 |
1 |
1 |
1 |
1 |
1 |
1 |
0.95 |
0.95 |
0.95 |
1 |
1 |
1 |
1 |
1 |
1 |
1 |
1 |
1 |
1 |
1 |
1 |
1 |
1 |
1 |
0.95 |
0.95 |
0.95 |
|
13 |
1 |
1 |
1 |
1 |
1 |
1 |
1 |
1 |
1 |
1 |
1 |
1 |
1 |
1 |
1 |
0.95 |
0.95 |
0.95 |
1 |
1 |
1 |
0.95 |
0.95 |
0.95 |
1 |
1 |
1 |
0.95 |
0.95 |
0.95 |
1 |
1 |
1 |
1 |
1 |
1 |
1 |
1 |
1 |
1 |
1 |
1 |
1 |
1 |
1 |
1 |
1 |
1 |
|
14 |
1 |
1 |
1 |
1 |
1 |
1 |
1 |
1 |
1 |
1 |
1 |
1 |
1 |
1 |
1 |
1 |
1 |
1 |
1 |
1 |
1 |
1 |
1 |
1 |
1 |
1 |
1 |
1 |
1 |
1 |
1 |
1 |
1 |
0.95 |
0.95 |
0.95 |
1 |
1 |
1 |
0.95 |
0.95 |
0.95 |
1 |
1 |
1 |
0.95 |
0.95 |
0.95 |
Remarks 9:For Table 3, "Comparison of optimization schemes," the authors need to provide the units for the values.
Response 9
Yes, we have made a modification to the article. The final value of the objective function in $/h is indicated in the first two columns of the table, while the latter two columns indicate the number of iterations required to achieve stabilization.
Table 3. Comparison of optimization schemes.
|
Method 1 Optimal Value ($/h) |
Method 2 Optimal Value ($/h) |
Method 1 Optimization Algebra |
Method 2 Optimizing Algebra |
|
|
1st |
2205.46 |
2203.83 |
938 |
816 |
|
2nd |
2204.63 |
2205.20 |
922 |
804 |
|
3rd |
2203.65 |
2204.96 |
929 |
803 |
|
4th |
2205.14 |
2205.50 |
1049 |
821 |
|
5th |
2203.71 |
2205.44 |
1055 |
862 |
|
6th |
2203.87 |
2203.91 |
952 |
788 |
|
7th |
2204.96 |
2203.99 |
1056 |
804 |
|
8th |
2203.65 |
2204.20 |
920 |
862 |
|
9th |
2204.88 |
2204.03 |
1060 |
815 |
|
10th |
2204.45 |
2204.73 |
964 |
754 |
|
Average value |
984.4 |
812.9 |
||
Remarks 10:The reviewer cannot fully understand the meaning of the vertical axis unit in relation to the convergence iteration value of the objective function.
Response 10
The vertical axis represents the minimum value of the objective function in every generation. It denotes the lowest value achieved in each iteration of the genetic algorithm, which is computed from a minimum of 200 individuals. The population comprises 400 individuals in each generation, out of which 200 are parents and 200 are offspring from the previous generation(14 offspring were generated through our Gene-Directed-Change method).
We hope our response meets your expectations. Yours advice are very helpful, thank you for your advice again. Please do not hesitate to let us know if you have any further suggestions in the future.
Kind Regards,
Yours sincerely,
Junjun Zhuang

Round 2
Reviewer 1 Report (New Reviewer)
Previous comments from the review have been successfully considered and the paper deserves to be published.
Reviewer 2 Report (New Reviewer)
All my comments and suggestions have been correclty addressed and addes in the revsied work. Now the paper is ready for the publication in this form by having increased its quality.
Reviewer 3 Report (New Reviewer)
The revised manuscript can be accepted after the significant revisions.
This manuscript is a resubmission of an earlier submission. The following is a list of the peer review reports and author responses from that submission.
Round 1
Reviewer 1 Report
Dear authors,
I am not critic with your work, I know about Wind Tunnels, but not so much about the cooling system. But I think it should not be published:
1. Firstly, is not clear any innovation in this work, so I am not sure that this paper should be published in a scientific journal. Genetic Algorithms have been used for decades and this is a design application.
2.There is not a physical explanation of the problem, so the reader does not know what are the different variables and parameters. There is not any innovation in the physical design of the problem.
3. There is not a context. Why is so important to control the enviroment in this kind Wind Tunnel? What are the parts of the WInd Tunnel?
Kind Regards,
Author Response
Dear reviewer:
Thank you very much for your hard work and the comments concerning our manuscript. We have read through comments carefully and have made corrections.
It is regrettable that this is not within your field. However, I am honored to introduce you to wind tunnel cooling systems. The cooling system is a crucial sub-system in wind tunnels, as during operation, wind tunnels generate substantial heat which the cooling system is designed to mitigate. Additionally, wind tunnel experiments require a lower temperature, necessitating the implementation of a refrigeration system to regulate the temperature of the experimental section. For your remaining comments, I would like to provide my response here.
Comment 1: Firstly, is not clear any innovation in this work, so I am not sure that this paper should be published in a scientific journal. Genetic Algorithms have been used for decades and this is a design application.
Response 1
We apologize for any confusion caused by our article's lack of focus.
While the genetic algorithm mentioned has been utilized since 1975, continuous enhancements have been implemented. Our contribution is the "Gene-Directed-Change Genetic Algorithm," which improves upon the existing algorithm. This is a new algorithm which utilizes fixed-point gene mutation technology, improving computational efficiency by approximately 17%. Moreover, the algorithm displays increased efficiency during the early computation stage compared to the traditional genetic algorithm.
Comment 2 :There is not a physical explanation of the problem, so the reader does not know what are the different variables and parameters. There is not any innovation in the physical design of the problem.
Response 2
I apologize for any confusion caused by our unclear presentation of variables. Allow me to provide a comprehensive description of our optimization variables, which fall under two distinct categories: system component selection and operating parameters.
The former pertains primarily to the centrifugal compressor, compressor, and expander. The characterization of each component is a crucial design parameter for such machines and is determined by the percentage of enthalpy drop of the vapour across the mobile vanes as compared to that of the first stage. The characterization of each component is a crucial design parameter for such machines and is determined by the percentage of enthalpy drop of the vapour across the mobile vanes as compared to that of the first stage. Additionally, a scaling factor indicates the size of the equipment relatively.
The operating parameter is made up of three main parameters; one of which is the relative speed of the centrifugal compressor, elucidating its operating conditions. Two essential terms in this context are the mass flow of compressed air, which refers to the amount of compressed air used, and the mass flow rate of cooling water, which describes the cooling process of a water cooler.
Comment 3 There is not a context. Why is so important to control the environment in this kind Wind Tunnel? What are the parts of the Wind Tunnel?
Response 3
I am sorry for our initial introduction may have lacked clarity. Allow me to provide a revised introduction highlighting the importance of our thesis research. As a vital subsystem in the operation of a wind tunnel, the refrigeration system maintains the tunnel’s heat balance and consumes a significant amount of energy. It consists of several components, including suction compressors, turbo-compressors, water coolers, return coolers, and turbo-expanders. The coordination of these components and the appropriate strategy for their use determine the advantages and disadvantages of the refrigeration system. Therefore, optimizing the design of the refrigeration system can effectively enhance its operating efficiency and reduce costs. Our article provides a more detailed examination of these points. The cost of cooling the system decreased from $3929.6 per hour to $2204.814 per hour, resulting in a cost savings of approximately 43%.
The primary emphasis of our paper is on the cooling system of the wind tunnel, rather than the wind tunnel itself, as it is a subsystem within. The cooling system of a high-speed-train environmental wind tunnel is based on the Brayton inverse cycle and comprises specific components, as shown in Figure 1.The system follows the Brayton cycle in which air is compressed and expanded to generate low-temperature cooling air for the high-speed-train environmental wind tunnel. The main components include the inlet compressor, turbo-compressor, turbo-expander, water cooler, return cooler, and cooling tower.
Figure 1. Air compression refrigeration cycle
Thank you for your advice. Please do not hesitate to let us know if you have any further suggestions in the future.
Kind Regards,
Yours sincerely,
Zhuang Junjun

Reviewer 2 Report
REVIEW:
The manuscript “Optimizing the Cooling System of High-Speed-Train Environmental Wind Tunnels using the Gene-Directed-Change Genetic Algorithm” sent to Entropy presents an optimization model to design and operate the cooling system of wind tunnels and genetic algorithms to solve the problem. One algorithm is new for this problem. Very promising results are found both in objective values and in computation time.
.
Due to academic and industrial interest, the subject is in the scope of Entropy.
The manuscript is decently well written, regarding both structure and language. There are some typos and importantly one important equation is only half seen. The motivation and objective are clearly stated in the Introduction. The literature review could be more comprehensive. The model is not presented in a way that another person could reproduce the model based on the manuscript. The algorithms are presented well. The major results have been presented well. The conclusions are based on the results.
The clear novelty is in the new genetic optimization algorithm using fixed-point gene mutations.
I feel that this manuscript has clear potential to be published in Entropy, but I have some remarks that the authors could take into consideration:
· You could inform how many of these wind tunnels are built in China and globally, to understand the importance of solving the problem.
· Correct the typos
· The equation in page 5 is shown only partly.
· Rewrite the model in a way that someone else could code the model.
· A symbols list could be beneficial.
FINAL RECOMMENDATION from REVIEWER: MAJOR REVISION
Check the typos
Author Response
Dear reviewer:
Thank you for acknowledging our dissertation. Your feedback on our paper was invaluable in improving its quality. We appreciate your assistance in this matter.
We have carefully considered your comments and made the necessary changes to the essay accordingly.
Remarks 1: You could inform how many of these wind tunnels are built in China and globally, to understand the importance of solving the problem.
Response 1
This suggestion was beneficial, resulting in the revision of the first paragraph of our introduction. We included the significance of introducing the train environment wind tunnel in the original text:
High-speed railways play an increasingly important role in economic transport. As an essential experimental facility for the research and development of high-speed railways, the high-speed-railway environmental wind tunnel has received significant attention. Nowadays, some of the world's famous environmental wind tunnels for trains include the Climatic Wind Tunnel Vienna, the CIRA Icing Wind Tunnel, the McKinley Climatic Laboratory, and the Jules Verne climatic wind tunnel. They have played a significant role in advancing high-speed train technology. One notable example is the Climatic Wind Tunnel Vienna, which boasts an impeccable weather simulation system. China, the world's foremost high-speed rail nation, is yet to possess its own high-speed rail wind tunnels. Since the foundation of The National Innovation Centre of High-Speed Train in 2019, China has been seeking to construct its own environmental wind tunnels. However, constructing and operating high-speed-railway environmental wind tunnels, especially full-size ones, incurs high costs. As a vital subsystem in the operation of a wind tunnel, the refrigeration system maintains the tunnel’s heat balance and consumes a significant amount of energy. It consists of several components, including suction compressors, turbo-compressors, water coolers, return coolers, and turbo-expanders. The coordination of these components and the appropriate strategy for their use determine the advantages and disadvantages of the refrigeration system. Therefore, optimizing the design of the refrigeration system can effectively enhance its operating efficiency and reduce costs.
Remarks 2 :Correct the typos
Response 2
We really apologize for the typos in our article. We have thoroughly reviewed the paper, identified and corrected the errors. However, if there are any remaining typos, please inform us and we will gladly make corrections.
Remarks 3 :The equation in page 5 is shown only partly.
Response 3
We really sorry for the error made. We have reviewed and amended the formula, and also verified the accuracy of other formulas to prevent similar occurrences.
Remarks 4: Rewrite the model in a way that someone else could code the model.
Response 4
We've rewritten the method:
We model the operation of the refrigeration system in order to calculate its performance in various states. In the refrigeration system model, the turbine and compressor are analyzed using one-dimensional flow characteristic analysis and the principle of similarity. The recooler is modeled using the effective number of heat transfer units and the mean temperature difference design method classical approach. The key system components and operating parameters are then optimized in turn to obtain the exergoeconomic cost. The objective function is obtained by multiplying the result with the frequency coefficients of the wind tunnel and adding the effect on the other parameters. By multiplying the outcome by the wind tunnel's frequency coefficients and including the impacts on other parameters, the objective function may be derived.
In fact, this issue concerns us greatly. We would like to provide a more detailed description of the model, but doing so could detract from the focus of our paper. Instead, we will emphasize our new evaluation function and genetic algorithm in this thesis. In our previous article, “Designing an Environmental Wind Tunnel Cooling System for High-Speed Trains with Air Compression Cooling and a Sensitivity Analysis of Design Parameters”, we provided an extensive account of the methodology for the model. We also performed a sensitivity analysis of the crucial parameters, ultimately identifying the variables that warrant optimization.
Remarks 5 : A symbols list could be beneficial.
Response 5
We fully appreciate this suggestion and have included the symbols list at the beginning of the article, and moreover we have thoroughly reviewed and clarified the symbols in our formula. Thank you very much for your suggestion.
|
symbols list |
subscript variable symbol: |
||
|
k |
comment |
||
|
C |
exergoeconomic cost |
sys |
system |
|
E |
exergoeconomic cost change |
cyc |
refrigerating system |
|
F |
exergoeconomic cost with frequency coefficient |
h |
high temperature |
|
G |
exergoeconomic cost with an Experimental Cycle |
c |
normal temperature |
|
T |
frequency of environmental wind tunnel |
hs |
high speed |
|
Z |
construction cost of this system |
ms |
medium speed |
|
q |
cooling capacity |
ls |
low speed |
|
efficiency |
os |
the external impact |
|
|
shi |
actual |
||
|
xu |
requirement |
||
Thank you for your advice again. Please do not hesitate to let us know if you have any further suggestions in the future.
Kind Regards,
Yours sincerely,
Junjun Zhuang

Reviewer 3 Report
Titled "Optimizing the Cooling System of High-Speed-Train Environmental Wind Tunnels using the Gene-Directed-Change Genetic Algorithm” the paper addresses to an exergoeconomic evaluation wind tunnel refrigeration system and as it states “optimizing the design of the refrigeration system can effectively enhance its operating efficiency and reduce costs” when using two GA optimization schemes and 3 different system options. As a result of the research, it is expected cost saving of around 1724.78USD/h due to improved performance even if inferior construction costs when comparing different options. The disadvantage of the article is that the representation of the system causes confusion because Fig. 1 components such as “compactor”, “wind pump”, “a circle with no label” not described in the article, and there is also a lack of explanatory text between the system components and GA optimization applications, which would allow for a better understanding the exploitation and dimensioning of the cooling system or its components.
Comments:
What do the rows and columns show in figure (table!) 10?
How do the authors explain the results in Table 3? Are there notable differences among the presented values?
For what reason are the results of the iterations presented twice in figure 11 and 12?
Author Response
Dear reviewer:
Thank you for acknowledging our dissertation. Your feedback on our paper was invaluable in improving its quality. We appreciate your assistance in this matter.
We have carefully considered your comments and made the necessary changes to the essay accordingly.
Comments 1: The disadvantage of the article is that the representation of the system causes confusion because Fig. 1 components such as “compactor”, “wind pump”, “a circle with no label” not described in the article, and there is also a lack of explanatory text between the system components
Response 1
This is satisfactory that you have highlighted an issue, and it is accurate that we failed to provide clarity in the article. Please accept our apologies. We have substituted the prior diagram with a fresh one.
Figure 1. Air compression refrigeration cycle.
First of all the “compactor” should be “compressor”, “A circle with no label” is a gas container whose main function is to stabilize the air pressure in the system. “wind pump” is “wind tunnel”. It's the cooling objective of our cooling system
Comments: 2 What do the rows and columns show in figure (table!) 10?
Response 2
Sorry for any inconvenience caused by our previous misrepresentation. It is a matrix with a size of 14 by 48 and the first column represents the row number. The matrix is essential to the gene-targeting mutation strategy as it is used to multiply the best individuals from the previous generation in the genetic algorithm to obtain new offspring. Due to its size, we are unable to display the matrix in its entire form and therefore, we can only provide it as a picture. We would like to clarify that this is the caption for the gene-targeting mutation strategy. The matrix is essential to the gene-targeting mutation strategy as it is used to multiply the best individuals from the previous generation in the genetic algorithm to obtain new offspring.
Comments:3 How do the authors explain the results in Table 3? Are there notable differences among the presented values?
Response 3
There are no significant variances between the optimal values. However,This demonstrates that our method does not affect computational accuracy when compared to the original method. However, when compared to the Optimization Algebra, our algorithm computes slightly faster. Moreover, our explanation of this table is located at the top of Table 3.
Comments:4 For what reason are the results of the iterations presented twice in figure 11 and 12?
Response 4
It is great to receive your remark. The two charts exhibit likeness, with the second portraying a more detailed view of the initial segment of the first chart. We have illustrated it this way as our enhancement is more detectable during the initial stages of the genetic algorithm. Thus, we drew the Figure 12 to describe it.
We hope our response meets your expectations. Yours advice are very helpful, thank you for your advice again. Please do not hesitate to let us know if you have any further suggestions in the future.
Kind Regards,
Yours sincerely,
Junjun Zhuang

Round 2
Reviewer 1 Report
Hi,
Like I said in the previous review I don't think this kind of paper should be published in this journal.
Kind regards,
Miguel
Reviewer 2 Report
I wasn't pleased that no rebuttal was provided and the changes made to the manuscript were not presented either, but based on my reading the manuscript has been improved based on my recommendation, so I suggest that this manuscript is ACCEPTED.